# Epimutations driven by RNAi or heterochromatin evoke transient antimicrobial drug resistance in pathogenic *Mucor* fungi

Ye-Eun Son, Carlos Pérez-Arques, Joseph Heitman *

Department of Molecular Genetics and Microbiology, Duke University Medical Center, Durham, North Carolina, United States of America

* heitm001@duke.edu

## Abstract

Antimicrobial resistance (AMR) is a global health threat emerging through microbe adaptation, driven by genetic variation, genome plasticity or epigenetic processes. In this study, we investigated how the *Mucor circinelloides* species complex adapts to the antifungal natural product FK506, which binds to FKBP12 and inhibits calcineurin-dependent hyphal growth. In *Mucor bainieri*, most FK506-resistant isolates (90%) were found to be unstable and transient, readily reverting to being drug sensitive when passaged without drug, and with no associated DNA mutations. In half of the isolates (50%), FK506-resistance was conferred by RNAi-dependent epimutation in which small interfering RNAs (siRNAs) silenced the *fkbA* encoding FKBP12 post-transcriptionally. In contrast, most of the remaining FK506-resistant isolates (40%) were found to have undergone heterochromatin-mediated silencing via H3K9 dimethylation, transcriptionally repressing *fkbA* and neighboring genes. In these heterochromatic epimutants, only minimal enrichment of siRNA to the *fkbA* locus was observed, but in three of the four examples, siRNA was significantly enriched at a locus distant from *fkbA*. A similar mechanism operates in *Mucor atramentarius*, where FK506 resistance was mediated by ectopic heterochromatin silencing of *fkbA* and associated genes with siRNA spreading across the region. Heterochromatin-mediated *fkbA* epimutants exhibited stability during in vivo infection, suggesting epimutation could impact pathogenesis. These findings reveal that antifungal resistance arising through distinct, transient epimutation pathways involving RNAi or heterochromatin, highlighting adaptive AMR strategies employed by ubiquitous eukaryotic microbes.

## Introduction

Fungi are ubiquitous eukaryotic organisms that occupy diverse ecological niches. While many fungi contribute to food production, pharmaceutical development, and biotechnology, others are pathogens capable of infecting a wide range of hosts

**Data availability statement:** All primary data are within the paper and its Supporting information files. Small RNA-seq, ChIP-seq, Nanopore sequencing data, and the corresponding genome assemblies used in this study are available from the NCBI SRA under project accession numbers PRJNA1242486 (*M. bainieri* CBS293.63) and PRJNA1242487 (*M. atramentarius* CBS202.28). Illumina whole-genome sequencing data are available under project accession numbers PRJNA1277536 (*M. bainieri* CBS293.63) and PRJNA1277544 (*M. atramentarius* CBS202.28). Numerical data for all graphs are provided in S1 Data.

**Funding:** This study was supported by the NIH/National Institute of Allergy and Infectious Diseases (https://www.niaid.nih.gov/; R01-AI170543-04, R01-AI039115-28, and R01-AI050113-20) awarded to J.H. J.H. is co-director and fellow of the CIFAR program Fungal Kingdom: Threats & Opportunities. The funders had no role in study design, data collection and analysis, decision to publish, or preparation of the manuscript.

**Competing interests:** I have read the journal's policy and the authors of this manuscript have the following competing interests: JH is a member of PLOS Biology's Editorial Board. The other authors declare that no competing interests exist.

**Abbreviations:** AMR, antimicrobial resistance; ChIP, chromatin immunoprecipitation; 5-FOA, 5-fluoroorotic acid; fAMR, fungal antimicrobial resistance; gDNA, genomic DNA; H3K9me2, histone 3 lysine 9 dimethylation; HMW, high-molecular-weight; MCC, *Mucor circinelloides* complex; PCR, polymerase chain reaction; PS, phylogenetic species; RITS, RNA-induced transcriptional silencing; RNAi, RNA interference; RT-qPCR, reverse transcription quantitative PCR; sRNA, small RNA; siRNAs, small interfering RNAs; WT, wild-type.

including humans [1]. The global burden of fungal infections has risen dramatically, now accounting for an estimated 2 million deaths annually [2,3]. Despite this growing threat, therapeutic options remain limited due to a restricted antifungal arsenal, inadequate diagnostics, and an alarming rise in antifungal resistance. The widespread use of antifungal agents in agriculture and healthcare, combined with environmental pressures such as climate change, has intensified selective pressure on fungal populations, driving the emergence of fungal antimicrobial resistance (fAMR) [4–7].

Fungi employ both genetic and epigenetic strategies to adapt to antifungal stress [8,9]. Genetic changes include point mutations (missense/nonsense) and insertions and deletions (indels) in drug target genes or drug transporter genes or their regulators, as well as large-scale chromosomal alterations such as copy number variation and aneuploidy. These changes confer resistance in several human fungal pathogens including *Candida albicans*, *Cryptococcus neoformans*, *Aspergillus fumigatus*, and *Mucor circinelloides* [10–13]. In contrast, epigenetic modifications enable nonmendelian inheritance of resistance without altering the underlying DNA sequence. These include processes such as DNA methylation, RNA interference (RNAi), and ectopic heterochromatin [14].

In *M. circinelloides*, RNAi-dependent silencing of *fkbA*—which encodes FKBP12, the cellular target of FK506 (tacrolimus)—confers transient FK506 resistance that reverts in the absence of drug pressure [15]. RNAi-mediated silencing of *pyrF* or *pyrG* prevents conversion of 5-fluoroorotic acid (5-FOA) into its toxic form, leading to transient 5-FOA resistance in *M. circinelloides* [16]. In addition, heterochromatin-driven gene silencing has been implicated in stress adaptation. In *Schizosaccharomyces pombe*, exposure to caffeine, an environmental stressor, selects for epimutations in which the induced formation of histone H3 lysine 9 dimethylation (H3K9me2)-enriched heterochromatin islands silence genes and promote survival without genetic mutation [17].

Mucormycosis is a life-threatening infection caused by Mucorales species including *Mucor* and *Rhizopus* [18]. Spores enter the host via inhalation, traumatic injury, or the rhino-orbital-cerebral route, often resulting in rapid dissemination, especially in immunocompromised individuals with conditions such as diabetes mellitus, malignancy, malnutrition, organ transplantation, or trauma [19,20]. The mortality rate of mucormycosis ranges from 50% to 90%, and incidence surged globally during the COVID-19 pandemic, with over 3,000 deaths reported in India alone [21–23]. Recognizing the severity of mucormycosis, the World Health Organization (WHO) recently designated Mucorales as a high-priority fungal group in the Fungal Priority Pathogens List to drive research and therapeutic development [9,24]. Given both the severity and widespread incidence of mucormycosis, a deeper understanding of intrinsic resistance and the potential for acquired antifungal resistance is urgently needed to guide proper and efficient therapeutic interventions.

The *Mucor circinelloides* complex (MCC) comprises at least 16 phylogenetic species (PS), 7 of which have been isolated from clinical cases of mucormycosis [25]. Prior studies in *M. lusitanicus* and *M. circinelloides* have identified both mendelian and epigenetic mechanisms of FK506 resistance, including RNAi-mediated silencing

of *fkbA* in the absence of mutations [15,26–28]. However, whether such epigenetic regulation is conserved across other MCC species—or whether alternative mechanisms exist—remains unclear. In this study, we demonstrate that antifungal FK506 resistance in MCC arises primarily through diverse genetic and epigenetic mechanisms, including either RNAi-mediated or heterochromatin-mediated epimutations that silence the *fkbA* gene, which encodes FKBP12, the target of FK506. Drug resistance conferred by both classes of epimutation is unstable, and readily reverts following passage in the absence of drug. In contrast, mutations in the *fkbA* gene confer stable resistance that is maintained even without drug pressure. Additionally, we demonstrate that FK506 resistance mediated by H3K9me2 heterochromatin is stably inherited after in vivo infection, providing new insights into the durability and clinical relevance of epigenetic antifungal resistance.

## Results

### Screening of *Mucor circinelloides* species complex (MCC) reveals isolates resistant to FK506

Previous studies reported the phylogenetic classification of the MCC and divided the group into 16 PS based on a polyphasic approach that included multi-locus sequence analyses, as well as morphological and physiological characterization [25]. RNAi-mediated epigenetic gene silencing has been shown to play a key role in FK506 resistance in *Mucor lusitanicus* (PS10) and *M. circinelloides* (PS14 and PS15) [15,26]. In this study, we screened multiple species to isolate strains resistant to FK506, a calcineurin inhibitor that prevents the dimorphic transition from yeast to hyphae, a process linked to virulence of *Mucor* [28], and rapamycin which also binds FKBP12 to form a complex that inhibits TOR and blocks nutrient-stimulated growth [29]. Following the strategy outlined in **Fig 1A**, spores were collected and point-inoculated on YPD media with or without FK506, and the emergence of resistant colonies was monitored. Resistant colonies were observed in *Mucor janssenii* (PS1), *Mucor bainieri* (PS3), and *Mucor atramentarius* (PS6), as well as in the previously reported species *M. lusitanicus* (PS10) and *M. circinelloides* (PS14 and PS15) (S1 Fig).

To investigate the underlying mechanisms of resistance, we selected FK506-resistant isolates from *M. janssenii*, *M. bainieri*, and *M. atramentarius* for further analysis (Fig 1B). The *fkbA* gene, three genes encoding calcineurin catalytic A (*cnaA*, *cnaB*, *cnaC*) subunits, and the calcineurin regulatory B subunit (*cnbR*) gene were amplified and analyzed by sequencing. As summarized (**Table 1**), four out of 10 FK506-resistant isolates from *M. janssenii* exhibited large deletions spanning the *fkbA* locus. An additional five isolates carried large insertions (~4–5 kb) in the *fkbA* promoter, nonsense mutations within *fkbA* exons, or point mutations at splice sites. These *fkbA* mutants were cross-resistant to rapamycin. In one isolate, a point mutation in the calcineurin A subunit gene (*cnaA*) was associated with FK506 resistance, but this isolate remained sensitive to rapamycin. Thus, in *M. janssenii*, all 10 isolates exhibited mendelian mutations. In contrast, analysis of 10 FK506-resistant isolates from *M. bainieri* revealed only one isolate with a nonsense mutation in *fkbA*, while the remaining nine exhibited no detectable sequence changes in *fkbA*, or the genes encoding calcineurin A or B subunits, suggesting a possible epigenetic mechanism (**Table 1**). Similarly, in *M. atramentarius*, among 10 FK506-resistant isolates, seven harbored various mutations in *fkbA* (3 nonsense mutations, 1 synonymous mutation, 2 splice-site point mutations, 1 start-loss mutation) and one had a missense mutation in the calcineurin B subunit (*cnbR*) (**Table 1**). Notably, two isolates showed no genetic alterations in any of the sequenced genes.

In summary, these results indicate that all 10 FK506-resistant isolates from *M. janssenii* are mendelian mutants. In *M. bainieri*, one isolate exhibited a mendelian mutation, while the remaining nine were identified as candidate epimutants. In *M. atramentarius*, eight isolates carried mendelian mutations, and two were classified as candidate epimutants.

### FK506-resistant epimutants display reversible phenotypes characterized by decreased levels of *fkbA* mRNA and FKBP12 protein expression

For FK506-resistant isolates where no DNA sequence alterations were detected in the FK506 target gene *fkbA* or the calcineurin genes, we hypothesized that resistance might be mediated by epigenetic mechanisms. Because epimutations are typically associated with reversible phenotypic plasticity [16], we tested whether FK506 resistance would be lost after

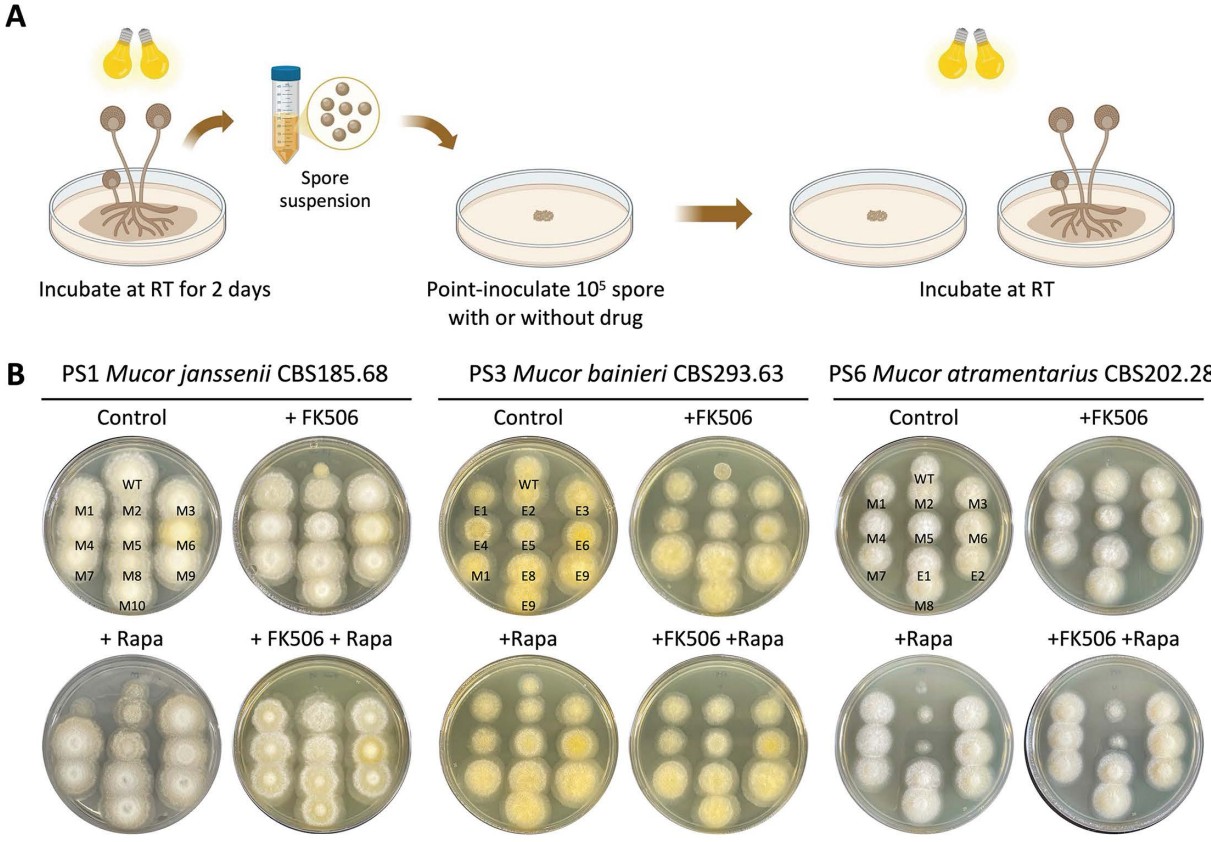

**Fig 1. FK506-resistant strains isolated from the *Mucor circinelloides* complex. (A)** Schematic representation of the procedure followed to obtain FK506- and/or rapamycin-resistant strains from *M. circinelloides* complex species. This figure was generated with BioRender.com. **(B)** Phenotypic analysis of FK506-resistant isolates in *Mucor janssenii* CBS185.68 (PS1), *Mucor bainieri* CBS293.63 (PS3), and *Mucor atramentarius* CBS202.28 (PS6). WT, wild-type; M, mendelian mutant; E, epimutant.

successive subculturing on drug-free medium. As expected, FK506 resistance was progressively lost following serial passages in the absence of selective pressure (**Fig 2A and 2D** and **Table 2**). Resistant strains exhibited a hyphal morphology and were capable of sporulation on FK506-containing media, whereas both wild-type (WT) and revertant strains displayed a yeast-like morphology when exposed to FK506 (**S2 Fig**). Additionally, in *M. bainieri*, phenotypic reversion occurred after an average of 15.5 passages, whereas in *M. atramentarius*, reversion required approximately 30 passages (**Fig 2A** and **2D**). Next, considering that epigenetic alterations inhibit target gene expression without changes in the underlying nucleotide sequence, we assayed *fkbA* transcript and FKBP12 protein levels. As shown in **Fig 2B** and **2E**, the *fkbA* mRNA levels in all FK506-resistant strains were significantly reduced compared to WT and were restored in all reverted strains. Consistently, FKBP12 protein was undetectable in resistant strains and was restored in reverted epimutant isolates (**Fig 2C** and **2F**). These results suggest that drug-resistant strains lacking genetic changes represent epimutants, and that resistance reversion varies both between and within species.

## Small RNAs confer FK506 resistance in *M. bainieri*

To determine which epigenetic mechanisms are involved in FK506 resistance in *M. bainieri*, we first evaluated the potential role of 5-methylcytosine (5mC) DNA methylation. Whole-genome sequencing data revealed the absence of canonical

**Table 1. FK506-resistant strains isolated in this study.**

| Species | Name | Phenotype | *fkbA* mutation | Impact on FKBP12 | Calcineurin mutation |
|---|---|---|---|---|---|
| PS1 *Mucor janssenii* | M1 | FK506-R, Rapa-S | No | No | G1496C in *cnaA* (G347A missense mutation) |
| | M2 | FK506-R, Rapa-R | 4,324 bp insertion (Unknown) at 38 bp upstream *fkbA* ORF | – | – |
| | M3 | FK506-R, Rapa-R | Large deletion | Loss | – |
| | M4 | FK506-R, Rapa-R | Large deletion | Loss | – |
| | M5 | FK506-R, Rapa-R | 5,052 bp insertion (RC/Helitron_DNA transposon) at 16 bp upstream *fkbA* ORF | – | – |
| | M6 | FK506-R, Rapa-R | C120 1 bp deletion in exon 2 (Frameshift) | Nonsense mutation | – |
| | M7 | FK506-R, Rapa-R | Large deletion | Loss | – |
| | M8 | FK506-R, Rapa-R | G258C in 5′ splice site of intron 2 | – | – |
| | M9 | FK506-R, Rapa-R | Large deletion | Loss | – |
| | M10 | FK506-R, Rapa-R | G258C in 5′ splice site of intron 2 | – | – |
| PS3 *Mucor bainieri* | E1 | FK506-R, Rapa-R | No | No | No |
| | E2 | FK506-R, Rapa-R | No | No | No |
| | E3 | FK506-R, Rapa-R | No | No | No |
| | E4 | FK506-R, Rapa-R | No | No | No |
| | E5 | FK506-R, Rapa-R | No | No | No |
| | E6 | FK506-R, Rapa-R | No | No | No |
| | M1 | FK506-R, Rapa-R | C357 1 bp deletion in exon 3 (Frameshift) | Nonsense mutation | No |
| | E7 | FK506-R, Rapa-R | No | No | No |
| | E8 | FK506-R, Rapa-R | No | No | No |
| | E9 | FK506-R, Rapa-R | No | No | No |
| PS6 *Mucor atramentarius* | M1 | FK506-R, Rapa-R | C404 1 bp deletion in exon 3 (Frameshift) | Nonsense mutation | No |
| | M2 | FK506-R, Rapa-S | No | No | G477T in *cnbR* (V122F missense mutation) |
| | M3 | FK506-R, Rapa-R | G249A in 5′ splice site of intron 2 | No | No |
| | M4 | FK506-R, Rapa-R | G249A in 5′ splice site of intron 2 | No | No |
| | M5 | FK506-R, Rapa-S | G189T in exon 2 | Synonymous mutation | No |
| | M6 | FK506-R, Rapa-R | C404 1 bp deletion in exon 3 (Frameshift) | Nonsense mutation | No |
| | M7 | FK506-R, Rapa-R | T2C in exon 1 (start-loss mutation) | Loss | No |
| | E1 | FK506-R, Rapa-R | No | No | No |
| | E2 | FK506-R, Rapa-R | No | No | No |
| | M8 | FK506-R, Rapa-R | C404 1 bp deletion in exon 3 (Frameshift) | Nonsense mutation | No |

5mC DNA 5-methylcytosine methyltransferases (DNMT1, DIM-2, RID-1, DNMT4, and DNMT5), consistent with a previous study (S3A Fig) [30]. Furthermore, 5mC analysis via Nanopore sequencing in the two *M. bainieri* wild-type strain detected no cytosine methylation, while a clear 5mC signal was detected in a *C. neoformans* clinical isolate with the same analysis pipeline [31], suggesting that this mechanism is not involved in FK506 resistance (S3B Fig).

We next examined the involvement of RNAi by performing small RNA (sRNA) sequencing. As shown in Figs 3A and S4, antisense sRNAs mapping to the *fkbA* locus were detected in five of the nine FK506-resistant epimutants but were absent in WT and revertant strains. The distribution pattern of sRNAs varied across epimutants. In epimutants E5–E9, *fkbA*-specific sRNAs strongly accumulated to ~1,000 RPM, whereas in WT and most revertants, no reads were detected, with only a minimal signal (~1 RPM) observed in E9 revertants (**Fig 3B**). The 5′ nucleotide of these sRNAs was predominantly

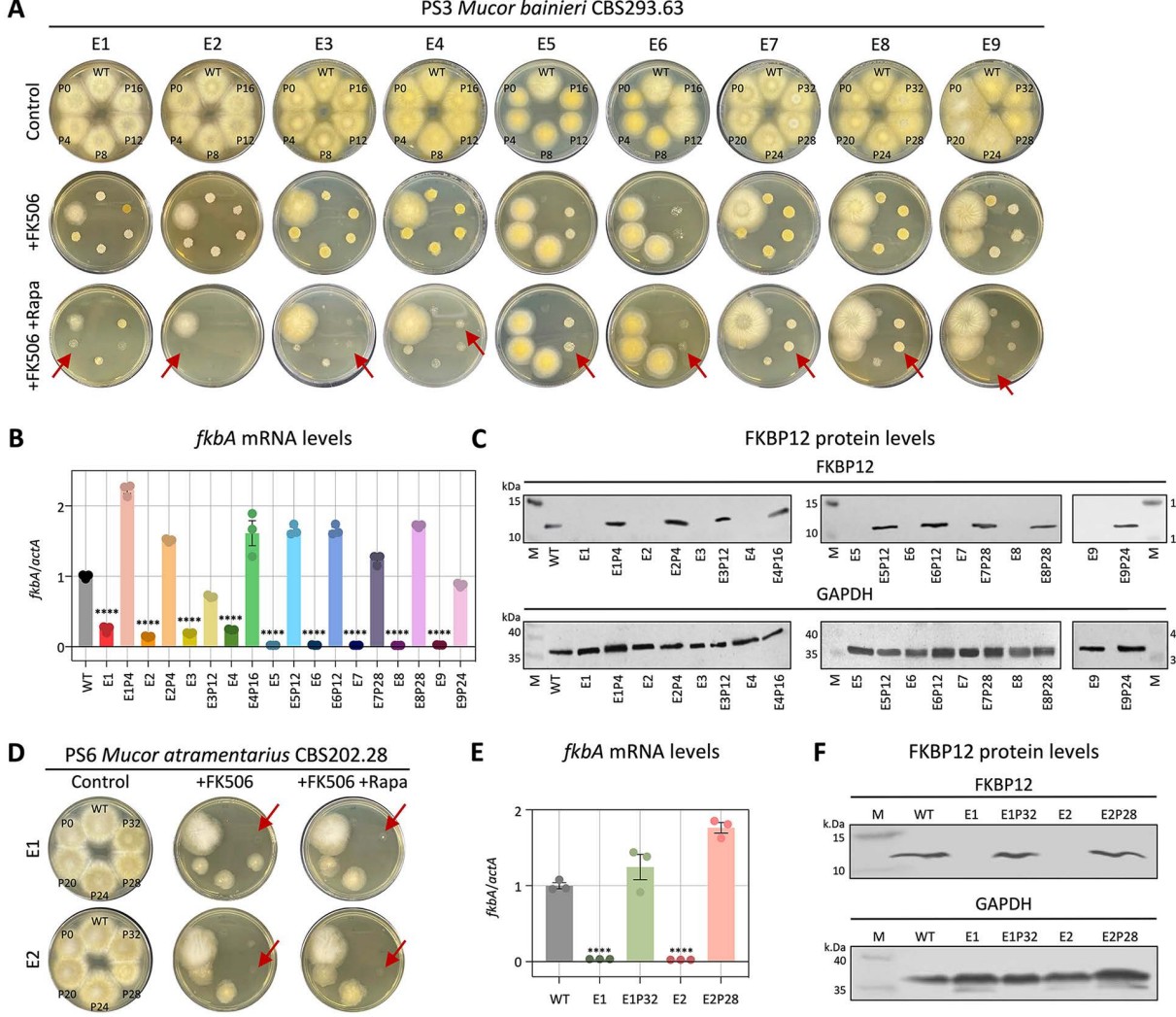

**Fig 2. FK506-resistant epimutants exhibit silencing of *fkbA* mRNA and FKBP12 protein expression and are unstable. (A, D)** Phenotypic reversion of FK506-resistant epimutant strains following serial passage without drug selection in *Mucor bainieri* **(A)** and *Mucor atramentarius* **(D)**. Red arrows indicate the points at which epimutants reverted to the WT phenotype. FK506 was utilized at 1 μg/mL and rapamycin at 100 ng/mL. WT, wild-type; P, number of passages. **(B, E)** Quantification of the *fkbA* mRNA expression in *M. bainieri* **(B)** and *M. atramentarius* **(E)**. Error bars represent mean ± SEM ($n = 3$). Statistical significance: $^{****}p \leq 0.0001$. **(C, F)** Western blot analysis of FKBP12 protein expression in *M. bainieri* **(C)** and *M. atramentarius* **(F)**. PS3: E1–E9, epimutants; E1P4–E9P24, revertants. PS6: E1 and E2, epimutants; E1P32 and E2P28, revertants. The data underlying this figure can be found in S1 Data.

uracil (~82%; **Fig 3C**), and their length distribution peaked at 21–24 nucleotides (nt) (**Fig 3D**), consistent with the general features of Argonaute-loaded sRNAs [15,32]. These results indicate that the E5–E9 epimutants are RNAi-dependent epimutants, in which FK506 resistance is mediated by *fkbA*-targeting antisense sRNAs that silence *fkbA* gene expression.

In contrast to epimutants E5–E9, which exhibited abundant small interfering RNAs (siRNAs) mapping to *fkbA*, epimutant isolates E1–E4 lacked detectable *fkbA*-targeting siRNAs (~1 RPM or less) (**Fig 3A** and **3B**). However, E1–E3 displayed siRNAs mapping to neighboring genes distant from *fkbA* (S5 Fig), with a strong 5′ uridine bias and predominantly 21–24 nt in length. For example, at *PS3_001333*, sRNAs reached ~2,595 RPM in epimutants E1-E3 but were largely absent in the WT (0.15 RPM) and revertants (1–1.6 RPM). Strikingly, epimutant E4 lacked detectable siRNAs at both

**Table 2. Epimutant reverted strains analyzed in this study.**

| Species | Name | Background | Phenotype | Description |
|---|---|---|---|---|
| **PS3 *Mucor bainieri*** | E1P4 | E1 | FK506-S, Rapa-S | E1 Revertant after passage 4 |
| | E2P4 | E2 | FK506-S, Rapa-S | E2 Revertant after passage 4 |
| | E3P12 | E3 | FK506-S, Rapa-S | E3 Revertant after passage 12 |
| | E3P16 | E4 | FK506-S, Rapa-S | E4 Revertant after passage 16 |
| | E5P12 | E5 | FK506-S, Rapa-S | E5 Revertant after passage 12 |
| | E6P12 | E6 | FK506-S, Rapa-S | E6 Revertant after passage 12 |
| | E7P28 | E7 | FK506-S, Rapa-S | E7 Revertant after passage 28 |
| | E8P28 | E8 | FK506-S, Rapa-S | E8 Revertant after passage 28 |
| | E9P24 | E9 | FK506-S, Rapa-S | E9 Revertant after passage 24 |
| **PS6 *Mucor atramentarius*** | E1P32 | E1 | FK506-S, Rapa-S | E1 Revertant after passage 32 |
| | E2P28 | E2 | FK506-S, Rapa-S | E2 Revertant after passage 28 |

*fkbA* and its neighboring loci. To determine whether epimutation spreading to *fkbA* from adjacent loci was associated with transcriptional repression, we performed reverse transcription quantitative PCR (RT-qPCR). As shown in S5K Fig, E1–E4 exhibited significant downregulation of genes flanking *fkbA*. However, no significant changes were observed in the expression of *fkbA*-neighboring genes in epimutants E7–E9. Meanwhile, several genes (e.g., *PS3_001332*, *PS3_001331*) showed significantly altered expression in epimutants E5–E6, but these alterations were not associated with sRNA abundance.

## Heterochromatin marks, along with RNAi, contribute to epigenetic resistance in *M. bainieri*

Small RNA spreading has been closely linked to heterochromatin formation and epigenetic gene silencing. Small RNAs targeting specific loci can initiate or reinforce histone modifications, particularly H3K9 methylation [17,33]. To determine whether histone-mediated epigenetic regulation contributes to FK506 resistance in *M. bainieri*, we conducted chromatin immunoprecipitation (ChIP) to assess the abundance of histone H3 lysine 9 dimethylation (H3K9me2) in the following epimutants: E1–E3, which exhibit sRNA spreading; E4, which lacks detectable sRNAs but shows transcriptional repression of *fkbA* and neighboring genes; and E8, a canonical RNAi-dependent, heterochromatin-independent *fkbA* epimutant. ChIP-qPCR was conducted across the *fkbA* locus, targeting the 5′ UTR, two exonic regions, and the 3′ UTR. H3K9me2 enrichment at the positive control region (an siRNA-enriched locus validated by H3K9me2 qPCR) was comparable across all strains (Fig 4A). H3K9me2 levels at the *fkbA* locus were markedly increased in epimutants E1–E4, with fold enrichment ranging from ~5-fold to over 100-fold relative to WT, although the magnitude varied across loci and strains. By contrast, H3K9me2 enrichment in E8 and its revertant E8P28 was negligible and comparable to WT. These results indicate that histone modification is associated with unstable antifungal drug resistance in some epimutants (E1–E4) but not others.

To further investigate heterochromatin-mediated gene silencing, we performed H3K9me2 ChIP-sequencing on the epimutant strains E3, E4, and E8. All strains exhibited similar patterns and levels of H3K9me2 and sRNA at positive control loci, supporting the specificity of the assay (Fig 4B). However, in E3 and E4, H3K9me2 signals extended across the *fkbA* locus from neighboring regions, suggesting heterochromatin spreading (Fig 4C). Notably, while E3 showed both H3K9me2 enrichment and sRNA accumulation (distal to *fkbA*), E4 displayed H3K9me2 enrichment without detectable sRNAs. These ectopic heterochromatic regions spanning *fkbA* were absent in the WT and revertant isolates. In contrast, the RNAi-dependent epimutant E8 exhibited strong sRNA accumulation specifically at the *fkbA* locus but lacked H3K9me2 enrichment. Taken together, these findings reveal three distinct types of epigenetic strategies underlying FK506 resistance in *M. bainieri* (Table 3): RNAi-associated heterochromatin-dependent silencing (type 1), heterochromatin-dependent silencing (type 2), and RNAi-dependent silencing (type 3). Interestingly, type 1 epimutants required fewer passages

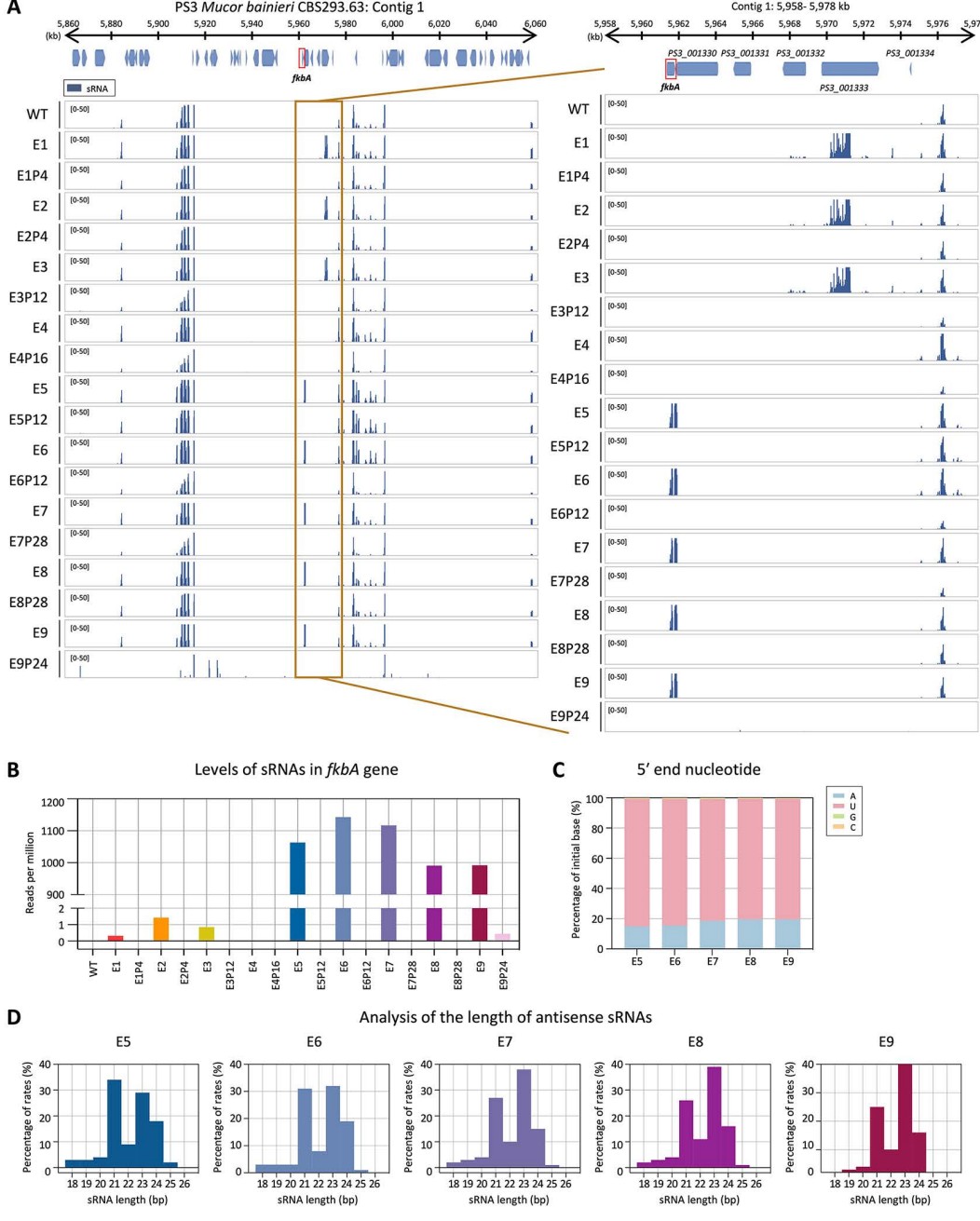

**Fig 3. Small RNAs mediate FK506 resistance in *Mucor bainieri*.** **(A)** Small RNA coverage mapped across the *fkbA* gene and its neighboring loci. The right panel presents a detailed view of the *fkbA* gene and its neighboring loci. Peaks shown below the coverage track indicate statistically enriched small RNA intervals identified by the MACS3 peak-calling algorithm, which models background signal to detect significant enrichment. **(B)** Small RNA abundance (RPM: reads per million) mapped to the *fkbA* gene. **(C)** 5′ end nucleotide preference of small RNAs associated with the *fkbA* gene. **(D)** Size distribution of small RNAs mapped to the *fkbA* gene. WT, wild-type; E1–E9, epimutants; E1P4–E9P24, revertants. The data underlying this figure can be found in S1 Data.

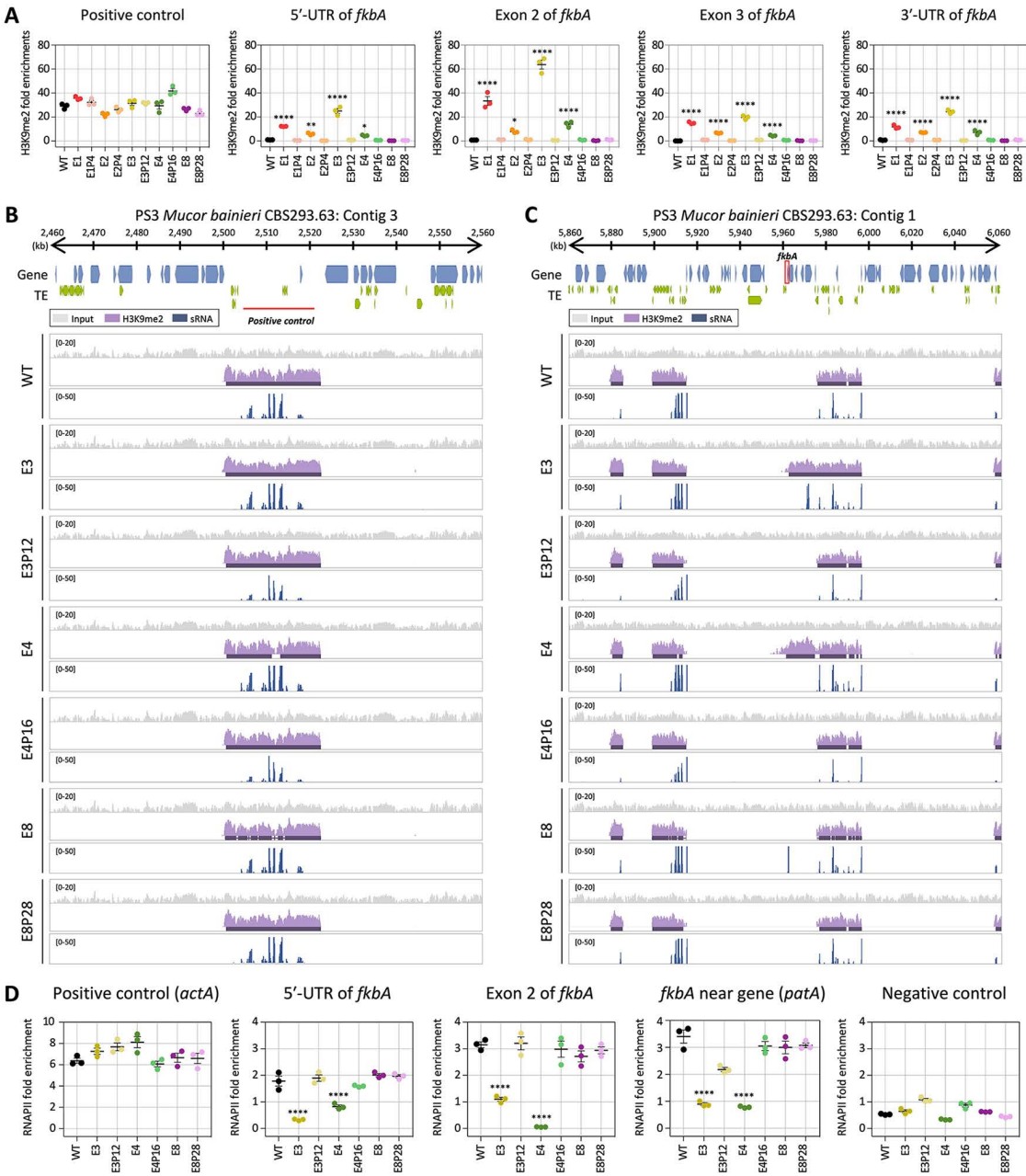

**Fig 4. Heterochromatin marks underlie epigenetic FK506 resistance in *Mucor bainieri*. (A)** ChIP-qPCR analysis of H3K9me2 enrichment at the positive control region (an siRNA-enriched locus validated by H3K9me2 qPCR) and *fkbA* locus (targeting the 5′ UTR, exonic regions, and 3′ UTR). Error bars represent mean±SEM ($n=3$). Statistical significance: *$p \le 0.05$, **$p \le 0.01$; ****$p \le 0.0001$. **(B, C)** ChIP-seq coverage mapped across the positive control locus (an siRNA-enriched locus validated by H3K9me2 qPCR; **B)** or *fkbA* and its neighboring loci **(C)**. The peaks shown beneath the coverage track denote significant H3K9me2-enriched intervals identified by the MACS3 peak-calling algorithm, which detects statistically enriched regions by modeling background signal. **(D)** ChIP-qPCR analysis of RNA polymerase II enrichment at the positive control region (*actA*), *fkbA* locus (targeting the 5′ UTR and exonic region), a *fkbA*-neighboring gene, and negative control (defined as an siRNA-producing region flanked by transposable elements and associated with H3K9me2). Statistical significance: ****$p \le 0.0001$. WT, wild-type; E1–E8, epimutants; E1P4–E8P28, revertants. The data underlying this figure can be found in S1 Data.

**Table 3. Summary of epigenetic mechanisms in *Mucor bainieri*.**

| Type | Mechanisms | Epimutant | Revertant | Average passages for revertant |
|------|-----------|-----------|-----------|-------------------------------|
| 1 | RNAi-associated heterochromatin-dependent silencing | E1 | E1P4 | 6.67 |
| | | E2 | E2P4 | |
| | | E3 | E3P12 | |
| 2 | Heterochromatin-dependent silencing | E4 | E3P16 | 16 |
| 3 | Canonical RNAi-dependent silencing | E5 | E5P12 | 20.8 |
| | | E6 | E6P12 | |
| | | E7 | E7P28 | |
| | | E8 | E8P28 | |
| | | E9 | E9P24 | |

(average of 6.67) for phenotypic reversion compared to epimutants in type 2 (16.00) or type 3 (20.80), suggesting that dual-layered epigenetic regulation may promote a more readily reversible resistance state.

Next, we sought to determine whether gene silencing in these epimutants is regulated at the pre- or post-transcriptional level. Generally, heterochromatin formation compacts chromatin structure, preventing RNA polymerase II (RNAPII) from accessing DNA and thereby blocking transcription initiation, which is classified as pre-transcriptional regulation [34]. In contrast, RNAi pathways typically involve sRNAs that target and degrade transcripts after transcription, representing post-transcriptional regulation [35]. To evaluate the transcriptional status of the *fkbA* locus, we performed RNAPII ChIP-qPCR assays. As shown in **Fig 4D**, RNAPII occupancy at a positive control region, *actA*, was comparable across all of the strains, while occupancy at the negative control region, which was enriched for sRNAs and marked by H3K9me2, was negligible in all strains. In contrast, the 5′ UTR, exon, and adjacent regions of *fkbA* showed significantly reduced RNAPII occupancy in E3 and E4 (where heterochromatin spreading was observed) compared to the WT, the RNAi-dependent epimutant E8, and revertant strains. These findings indicate that type 1 (RNAi-associated heterochromatin-dependent silencing) and type 2 (heterochromatin-dependent silencing) occurs at the pre-transcriptional level, whereas type 3 (RNAi-dependent silencing) operates exclusively through post-transcriptional regulation.

### Heterochromatin mediates FK506 resistance in *M. atramentarius*

In *M. atramentarius*, canonical 5mC DNA methyltransferases (DNMT1, DIM-2, RID-1, DNMT4, and DNMT5) were also absent, consistent with a previous study (S3A Fig) [30]. Moreover, 5mC analysis in the WT detected no cytosine methylation, suggesting that DNA methylation is not involved in FK506 resistance (S3B Fig). We next assessed whether RNAi- or heterochromatin-based resistance mechanisms identified in *M. bainieri* also operate in *M. atramentarius*, another member of the *Mucor circinelloides* species complex. To evaluate the potential involvement of RNAi, we performed sRNA sequencing of WT, two FK506-resistant epimutants, and their corresponding revertant strains. As shown in Fig 5A, antisense sRNA spreading was observed across the *fkbA* locus and its flanking regions, although sRNA abundance at *fkbA* itself was limited (S6 and S7B Figs). Overall the sRNAs exhibited a strong 5′ uridine bias (S7C Fig) and a length distribution peaking at 21–24 nt (S7D Fig), consistent with canonical RNAi-derived sRNAs. Interestingly, while the spreading pattern resembled that observed in heterochromatin-associated epimutants of *M. bainieri* (Figs 3A and 5A), *M. bainieri* accumulated markedly higher levels of sRNAs at genes flanking the *fkbA* locus compared to *M. atramentarius* (S4 and S6 Figs).

To further determine whether the resistant isolates of *M. atramentarius* also involve heterochromatin-associated mechanisms, we next examined the distribution of H3K9me2. ChIP-qPCR revealed significant H3K9me2 enrichment at the *fkbA* locus in the epimutants, with fold enrichment ranging from ~12-fold to over 60-fold relative to WT (**Fig 5B**). While H3K9me2 enrichment at the GremLINE1 transposon, used as a positive control, was comparable across all strains,

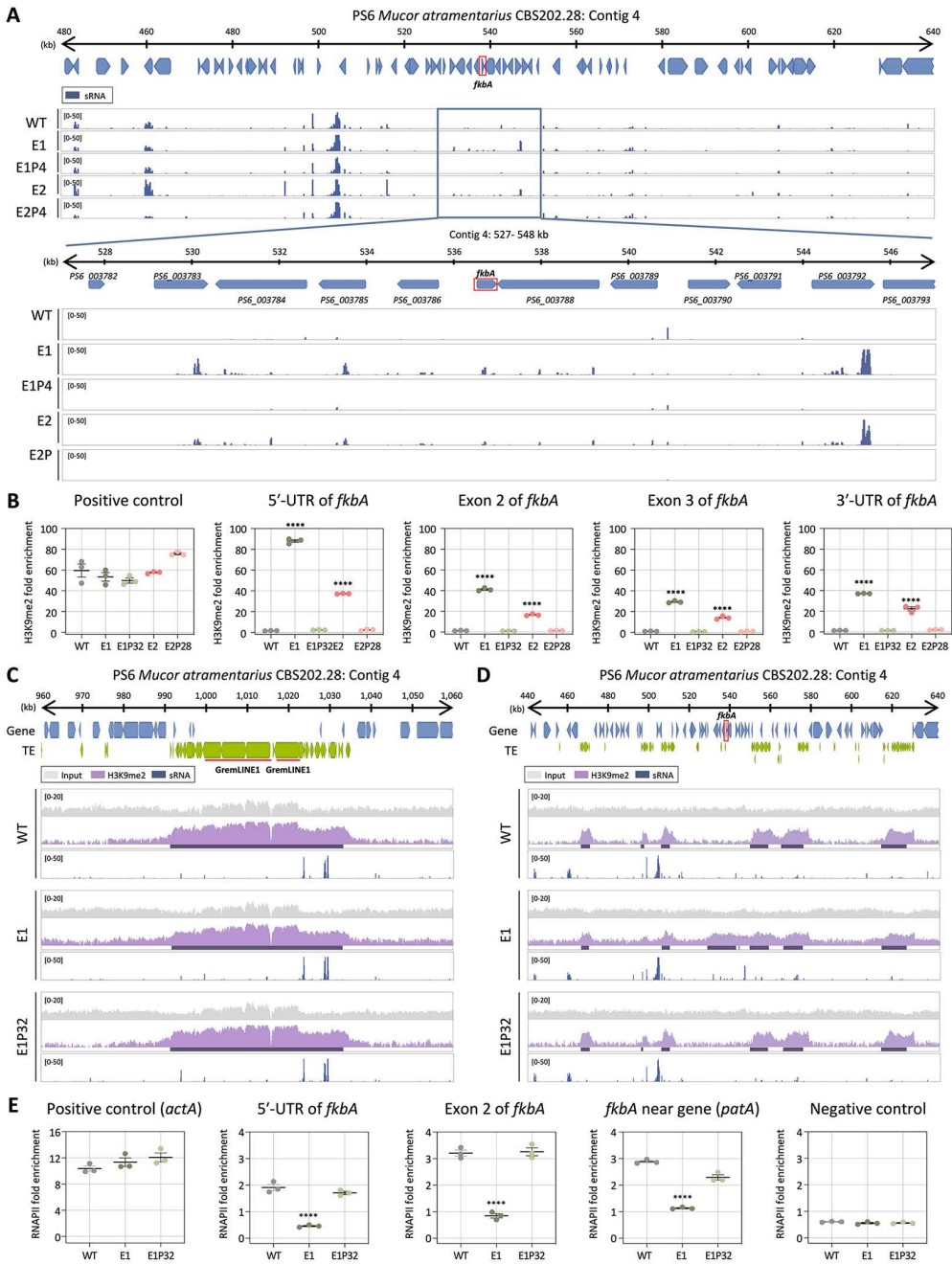

**Fig 5. Heterochromatin-associated FK506 resistance is conserved in _Mucor atramentarius_. (A)** Small RNA coverage mapped across the *fkbA* gene and its neighboring loci. Bottom panel presents a detailed view of the *fkbA* gene and its neighboring loci. Peaks shown below the coverage track indicate statistically enriched small RNA intervals identified by the MACS3 peak-calling algorithm, which models background signal to detect significant enrichment. **(B)** ChIP-qPCR analysis of H3K9me2 enrichment at the GremLINE1 transposon (positive control region) and *fkbA* locus (targeting the 5′ UTR, exonic regions, and 3′UTR). Error bars represent mean±SEM (*n* = 3). Statistical significance: ****$p \leq 0.0001$. **(C, D)** ChIP-seq coverage mapped across the GremLINE1 transposon and its neighboring loci **(C)** or *fkbA* and its neighboring loci **(D)**. The peaks shown beneath the coverage track denote significant H3K9me2-enriched intervals identified by the MACS3 peak-calling algorithm, which detects statistically enriched regions by modeling background signal. **(E)** ChIP-qPCR analysis of RNA polymerase II enrichment at the positive control region (*actA*), *fkbA* locus (targeting the 5′-UTR and exonic region), a *fkbA*-neighboring gene, and negative control locus (defined as an siRNA-producing region flanked by transposable elements and associated with H3K9me2). Statistical significance: ****$p \leq 0.0001$. WT, wild-type; E1 and E2, epimutants; E1P32 and E2P28, revertants. The data underlying this figure can be found in S1 Data.

expanded H3K9me2 signals at *fkbA* and its neighboring genes, accompanied by sRNA spreading, were strikingly observed exclusively in the epimutant (**Fig 5C** and **5D**). Consistently, RT-qPCR revealed significantly reduced expression of *fkbA*-neighboring genes in the epimutants compared to WT and revertant strains (S8 Fig). Furthermore, consistent with observations in *M. bainieri*, we detected a significant decrease in RNAPII enrichment at the 5′ UTR, exon, and neighboring regions of *fkbA* in the *M. atramentarius* epimutant, indicating that heterochromatin-mediated gene silencing in *M. atramentarius* operates pre-transcriptionally (**Fig 5E**). Collectively, these results support the conservation of heterochromatin-mediated FK506 resistance in *M. atramentarius*, reinforcing its role as a general epigenetic mechanism across the MCC.

### Antifungal resistance mediated by heterochromatin is stably inherited following in vivo infection

Previous studies reported that RNAi-mediated antifungal resistance can be stably transmitted following in vivo passage [36]. To investigate whether heterochromatin-mediated resistance also persists through cell division and stress during host infection, we first examined the maximum growth temperature of *M. bainieri* and *M. atramentarius* with *M. circinelloides* serving as a positive control. *M. atramentarius* was able to grow at 37°C, suggesting potential pathogenicity, whereas *M. bainieri* failed to grow at 37°C (**Fig 6A**). This was further confirmed in a murine model, as mice infected with *M. atramentarius* exhibited significantly reduced survival, comparable to that observed with *M. circinelloides*, a known pathogen that causes mucormycosis (**Fig 6B**).

Based on these findings, we infected immunosuppressed mice with equal numbers of spores from WT, FK506-resistant epimutant (E1), and revertant (E1P32) strains of *M. atramentarius*. On day 4 post-infection, brain, liver, spleen, and lung tissues were harvested to assess fungal dissemination. Fungal burden was comparable across strains, with no significant differences observed (S9A Fig). Subsequently, when fungal colonies recovered from each organ were tested on FK506-containing media, resistance was retained in colonies derived from epimutant-infected mice (over 80% across organs) in contrast to those derived from mice infected with WT and revertant strains (Fig 6C). Interestingly, a few colonies recovered from the lungs of WT- and revertant-infected mice also exhibited FK506 resistance. Sequencing of the *fkbA* locus in FK506-resistant isolates recovered from each organ of WT-, epimutant-, and revertant-infected mice revealed no mutations, with sequences identical to the WT, indicating that these isolates represent epimutants rather than mendelian mutants (S9B Fig and S3 Table). Given that heterochromatin mediates FK506 resistance in epimutants, we next asked whether this epigenetic state persists following in vivo infection. To address this, we performed ChIP-qPCR on fungal colonies recovered from four organs. FK506-resistant epimutants exhibited robust H3K9me2 enrichment not only at exonic regions but also across the *fkbA* promoter and terminator (Figs 6D and S10). By contrast, H3K9me2 levels were minimal or undetectable in WT and revertant strains. These findings indicate that heterochromatin-mediated antifungal resistance is largely stable during host infection, likely maintained through mitotic inheritance of H3K9 methylation marks.

To further examine whether epimutant stability persists over longer periods of infection, we extended our in vivo assays by administering a lower dose of cyclophosphamide (an immunosuppressant) or no cyclophosphamide treatment and then assessed FK506 resistance and H3K9me2-associated stability after two or four weeks. Unlike the results observed at 4 days post-infection, fungal burdens at 2 weeks post-infection with lower or no cyclophosphamide, and at 4 weeks post-infection without cyclophosphamide, were detected only in the liver and spleen, but not in the brain or lungs (S11A–S11C Fig). Next, when fungal colonies recovered from the liver and spleen were tested on FK506-containing medium, resistance was retained in colonies derived from epimutant-infected mice at frequencies greater than 50% under all conditions tested (Fig 7A, 7C, and 7E). We sequenced the *fkbA* locus in resistant colonies derived from the liver and spleen, confirming that there were no mutations, with sequences identical to the WT (S11D–S11F Fig and S4–S6 Tables). Furthermore, ChIP-qPCR analysis revealed that FK506-resistant epimutants exhibited H3K9me2 enrichment at the *fkbA* locus (Fig 7B, 7D, and 7F). Together, these results demonstrate that heterochromatin-mediated antifungal resistance in *M. atramentarius* is stably maintained during prolonged host infection, independent of genetic mutations, and is instead sustained by mitotically inherited H3K9me2 marks.

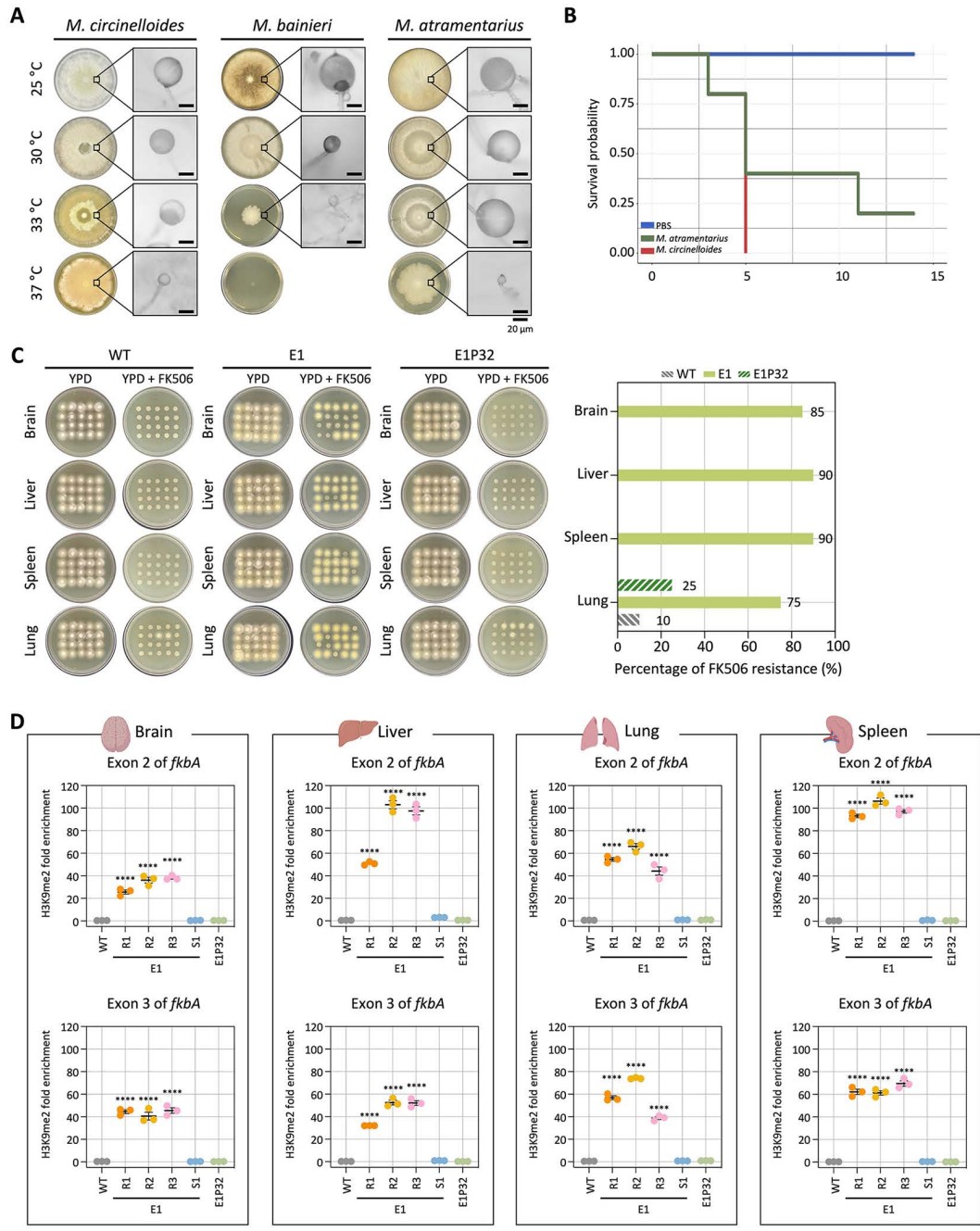

**Fig 6. Heterochromatin-mediated antifungal resistance persists after in vivo passage. (A)** Fungal growth of *Mucor circinelloides*, *Mucor bainieri* and *Mucor atramentarius* at different temperatures. Microscopic images show enlarged sporangia (scale bar = 20 μm). **(B)** Survival rates of mice infected with *M. atramentarius*. *M. circinelloides* and PBS served as positive and negative controls, respectively. The log-rank test revealed a statistically significant difference in survival between groups (*M. atramentarius* vs. PBS, $p = 0.014$). **(C)** Fungal colonies recovered from four different organs following in vivo infection with wild-type (WT), epimutant (E1), and revertant (E1P32) strains. The bar plot (right) shows the percentage of the colonies recovered after infection that are FK506-resistant. **(D)** ChIP-qPCR analysis of H3K9me2 enrichment at the *fkbA* locus following in vivo passage, targeting two exonic regions. Error bars represent mean ± SEM ($n = 3$). Statistical significance: ****$p \leq 0.0001$. WT, wild-type; E1R1–E1R3, FK506-resistant colonies; E1S1, FK506-sensitive colony; E1P32, in vitro revertant. The data underlying this figure can be found in S1 Data.

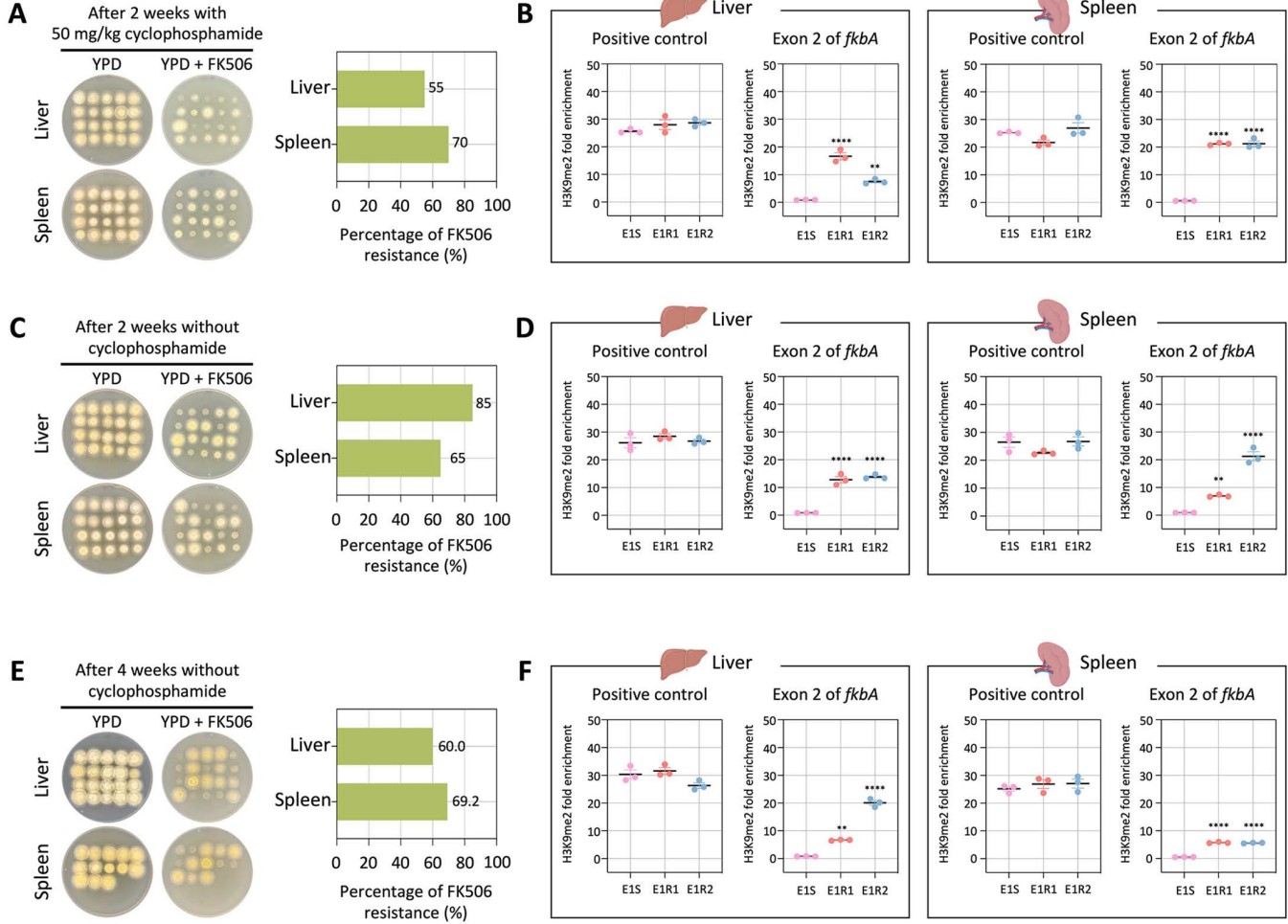

**Fig 7. Heterochromatin-mediated antifungal resistance remains stable after in vivo infection in immunosuppressed and non-immunosuppressed mice. (A−C)** Fungal colonies recovered from four different organs following in vivo infection with a heterochromatin-mediated epimutant under three different conditions: **(A)** cyclophosphamide-treated mice (50 mg/kg, 2 weeks post-infection), **(B)** untreated mice (2 weeks post-infection) and **(C)** untreated mice (4 weeks post-infection). The bar plots on the right show the percentage of FK506-resistant colonies recovered from different organs under each condition. **(D−F)** ChIP-qPCR analysis of H3K9me2 enrichment at the *fkbA* locus under the same three conditions: **(D)** cyclophosphamide-treated mice (50 mg/kg, 2 weeks post-infection), **(E)** untreated mice (2 weeks post-infection), and **(F)** untreated mice (4 weeks post-infection). Error bars represent mean ± SEM ($n = 3$). Statistical significance: $^{**}p \le 0.01$; $^{****}p \le 0.0001$. The data underlying this figure can be found in S1 Data.

## Discussion

fAMR represents an urgent and growing global health threat, jeopardizing human and animal health, agriculture, and food security [7]. Although antifungal drugs are widely used to mitigate this burden, the remarkable adaptability of fungi to chemical stressors has rendered them increasingly difficult to control. Paradoxically, dual antifungal therapies intended to improve treatment outcomes may instead accelerate the evolution of multidrug-resistant strains [37–39]. A deeper understanding of the mechanisms driving fAMR is critical for developing effective interventions across clinical and environmental settings.

FK506 targets FKBP12, a conserved protein that when bound to FK506 inhibits the calcineurin pathway—an essential regulator of fungal growth, morphogenesis, stress responses, and virulence [40]. In *M. circinelloides*, FK506 enforces a

morphological shift from hyphae to yeast cells, thereby attenuating pathogenicity [28]. Here, we show that FK506 regulates morphology across the MCC (S1 Fig), and that resistance arises via both mendelian and non-mendelian mechanisms (Fig 1 and Table 1). Our findings expand current models of antifungal resistance by revealing that RNAi- or heterochromatin-mediated epimutations represent alternative, and sometimes independent, pathways to resistance. These mechanisms may function separately or synergistically to silence *fkbA*, operating through either post-transcriptional or pre-transcriptional regulation to confer FK506 resistance, and resulting in resistance phenotypes with varying degrees of stability, specificity, and reversibility (Fig 8).

In nature, diverse microorganisms, including bacteria, viruses, and fungi, interact continuously, forming complex ecological networks [41–44]. Soil systems are nutrient-rich environments that support a vast diversity of microorganisms, which in turn contribute to essential ecological functions such as nutrient cycling and organic matter decomposition [45,46]. *Streptomyces* is one of the most widely distributed and ecologically significant genera within the phylum Actinobacteria, renowned for its remarkable ability to produce a wide variety of bioactive secondary metabolites, including antifungal compounds [47]. Here, we studied FK506, a natural antifungal compound that is produced by several *Streptomyces* species, such as *Streptomyces hygroscopicus* and *Streptomyces tsukubaensis* [48,49]. Mucorales represent a major fungal order of soil-dwelling saprophytic fungi [50], and thus the ecological coexistence of *Streptomyces* and *Mucor* in soil raises the possibility that natural occurrence of the antifungal natural product FK506 may impose a selective pressure on fungal populations. This pressure may promote the emergence of epigenetic resistance, even in the absence of anthropogenic drug use, by causing fungi to flexibly modulate gene expression in response to environmental antifungal stress. Interestingly, amphotericin B, the primary first-line therapy for mucormycosis, is also a natural product produced by *Streptomyces nodosus* [51]. This raises the possibility that fungi may encounter antifungal pressures in nature, allowing epigenetic resistance mechanisms to evolve as strategies that enhance survival and competitive fitness in chemically dynamic ecosystems.

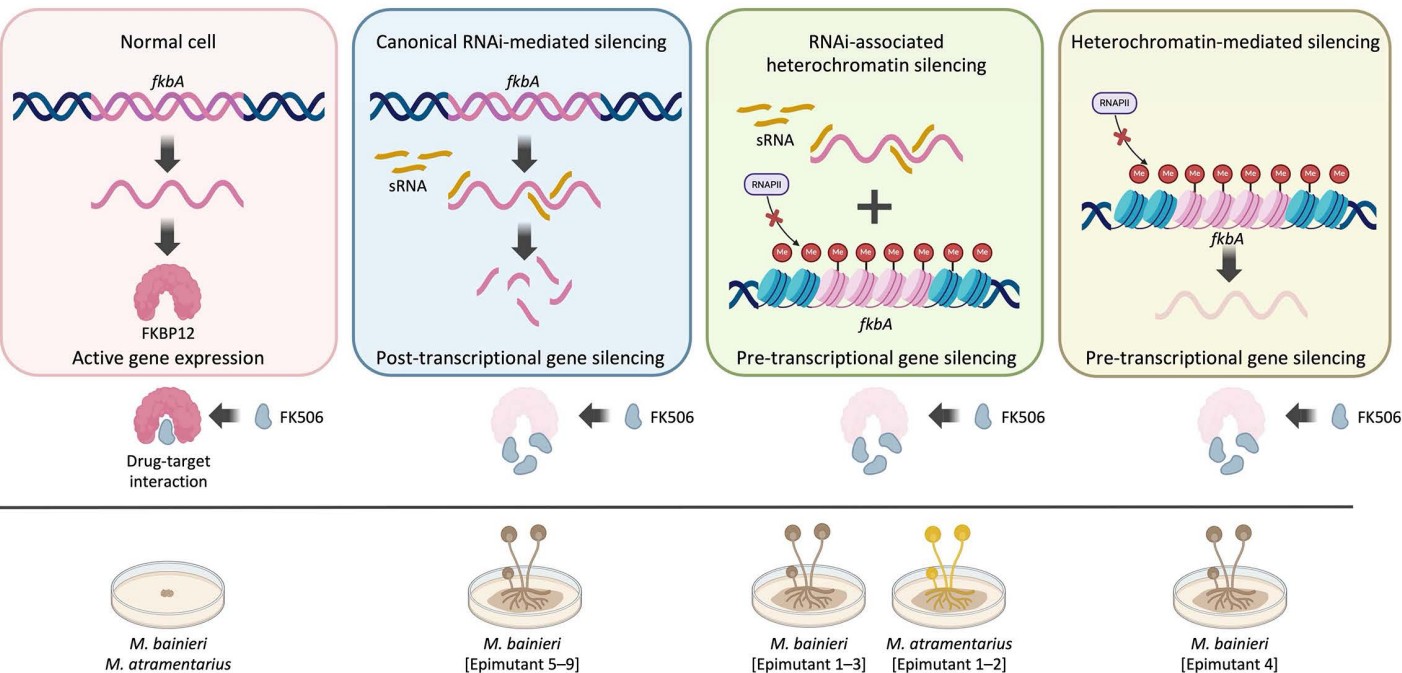

**Fig 8. Schematic overview of epigenetic silencing contributing to antifungal resistance in *Mucor* species.** *fkbA* expression enables FKBP12 production and FK506 sensitivity (yeast-like morphology). Epigenetic silencing of *fkbA* via RNAi and/or heterochromatin leads to FK506 resistance and hyphal growth. Generated with BioRender.com.

Our data revealed that the size distribution of sRNAs at the *fkbA* locus differs from those mapped to neighboring genes (Figs 3, S5, and S7). The predominant sRNAs derived from *fkbA* were 21 and 24 nt, whereas most neighboring genes displayed stronger accumulation at 23–24 nt. Previous work in *Mucor* proposed that different sRNA size classes may serve distinct roles in silencing, with 21-nt sRNAs acting as the predominant species involved in target mRNA degradation [52]. In this context, the increased abundance of 21 nt sRNAs derived specifically from *fkbA* suggests that FK506-resistant epimutants may preferentially generate the sRNA class most directly linked to efficient target degradation. A similar phenomenon was also reported at the *pyrG* and *pyrF* loci, where sRNAs fell within the 21–24 nt range but showed locus differences. *pyrG* sRNAs were enriched at 21 and 24 nt, whereas *pyrF* sRNAs were predominantly 23–24 nt [16]. Such variation may result from multiple, non-mutually exclusive mechanisms. First, differential usage of Dicer paralogs may contribute. In *M. circinelloides*, DCL2 produces 23–24 nt siRNAs, whereas DCL1 can result in more heterogeneous size distributions including 21 nt [53,54]. Similarly, in the model plant *Arabidopsis thaliana*, DCL4 primarily generates 21-nt siRNAs, while DCL3 is responsible for 24-nt siRNAs involved in transcriptional silencing [55,56]. Second, substrate properties of the transcripts, including secondary structure, GC content, and overall architecture, may influence Dicer processing efficiency and cleavage-site preference, thereby generating distinct size profiles [57,58]. These observations suggest that even within the canonical RNAi framework, locus-specific features may shape distinct sRNA size distributions.

The interplay between RNAi and histone methylation raises key questions about the sequence and causality of these silencing events. Preexisting heterochromatin can initiate RNAi, as seen in centromeres and transposable elements in *S. pombe* and *A. thaliana*, where transcripts from heterochromatic loci are processed into siRNAs that reinforce silencing [59]. In *S. pombe*, sRNAs bound to Argonaute form the RNA-induced transcriptional silencing (RITS) complex with the chromodomain protein Chp1 and the bridging protein Tas3. Chaperone proteins such as Hsp90 stabilize RITS complex, which recruits Clr4 to nascent transcripts, thereby facilitating heterochromatin formation [59,60]. Stc1 associates with Ago1 and recruits the chromatin-modifying CLRC complex (Clr4-Rik1-Cul4-Dos1-Dos2), underscoring its critical role in connecting RNAi with heterochromatin formation [61]. Meanwhile, euchromatic loci may sporadically generate double-stranded RNA, initiating RNAi and leading to heterochromatin deposition via a positive feedback loop involving the RITS complex and Clr4 [62]. Interestingly, in our study, we identified FK506-resistant epimutants with varying combinations of siRNA accumulation and H3K9me2 enrichment at the *fkbA* locus (**Fig 4**). These findings support several non-exclusive models: (1) RNAi-driven initiation of heterochromatin, (2) heterochromatin-facilitated RNAi, or (3) other mechanisms.

To explore the first hypothesis, we conducted a comparative analysis of RNAi, H3K9 methylation-based heterochromatin formation, and RITS components across *M. bainieri*, *M. atramentarius*, *M. lusitanicus*, and *M. circinelloides* to assess the potential coupling between RNAi and heterochromatin pathway. As reported previously in *M. lusitanicus* [30], the RITS components Chp1 and Tas3 are absent across all four *Mucor* species, whereas the bridging protein Stc1, the histone methyltransferase Clr4, and CLRC complex subunits (Rik1, Cul4, and Raf1) are conserved (S12 Fig). Thus, although elements of the coupling machinery such as Stc1 and CLRC are conserved, the apparent absence of a complete RITS complex suggests that the connection between RNAi and heterochromatin may operate in a partial or context-dependent manner. Under FK506 selective pressure, RNAi may be initiated at the *fkbA* locus, leading in some cases to the establishment of heterochromatin, while in others, RNAi may act independently. Even when heterochromatin is formed, weak or transient interactions between RNAi and heterochromatin could result in heterochromatin-only epimutants.

Second, preexisting heterochromatin could facilitate RNAi, serving as a template for sRNA generation. To explore this possibility, we analyzed the distribution of H3K9 dimethylation and sRNA in WT strains of four *Mucor* species (*M. bainieri*, *M. atramentarius*, *M. lusitanicus*, and *M. circinelloides*). As shown in S13 Fig, discrete H3K9me2 islands were detected downstream of *fkbA* in all four species, suggesting that preexisting heterochromatin on transposon-rich regions may spread toward the *fkbA* locus under FK506 selective pressure and recruit the RNAi machinery to generate sRNAs. Similar phenomena have been reported in other organisms. In *S. pombe*, for example, caffeine stress triggers redistribution of

H3K9 methylation and evokes sRNA generation [17]. Interestingly, these H3K9me2 islands lacked detectable sRNAs in *M. lusitanicus* and *M. circinelloides*, whereas the co-occurrence of H3K9me2 and sRNAs was observed in *M. bainieri* and *M. atramentarius*, raising the possibility that the relationship between RNAi and heterochromatin formation may interact differently across species.

Third, RNAi and heterochromatin formation may occur simultaneously under FK506 pressure but act independently without functional coupling. A previous study reported that RNAi and heterochromatin formation can function independently to repress GremLINE1 transposable elements and maintain genome stability in *M. lusitanicus* [30]. Thus, it is possible that, upon FK506 exposure, these machineries were independently activated at the same genomic region, with RNAi generating sRNAs at the *fkbA* locus and heterochromatin spreading from adjacent transposon-rich regions, resulting in their apparent coexistence without direct mechanistic interaction. Taken together, future studies analyzing pathway-specific mutants will be critical to determine the directionality and mechanistic coupling or uncoupling of these epigenetic processes.

Beyond FK506-induced dynamics at the *fkbA* locus, our comparative genomic analyses across four *Mucor* species provide additional insight into how RNAi and heterochromatin interplay may differ among species. GremLINE elements are present in *M. atramentarius*, *M. lusitanicus*, and *M. circinelloides*, where they reside within or near putative centromeric regions, but are absent from the *M. bainieri* genome (S14 Fig). The centromeric GremLINEs in these species consistently show strong H3K9me2 enrichment, supporting their association with heterochromatic domains. Notably, *M. bainieri* exhibits a distinct architecture. Although it lacks GremLINEs, its putative centromeric regions are enriched for a different repeat, TE00000158, which is associated with strong H3K9me2. This pattern diverges sharply from the GremLINE-associated uncoupling observed in *M. circinelloides*, raising the possibility that *M. bainieri* may have evolved a species-specific centromeric repeat landscape and RNAi–heterochromatin relationship. Collectively, these comparative observations suggest that the interplay between RNAi and heterochromatin may not be conserved and may vary across species, likely influenced by species-specific differences in repeat landscapes and centromere architecture.

Torres-Garcia and colleagues demonstrated that external stressors such as caffeine induce phenotypic plasticity via Clr4/H3K9me "read-write" systems in *S. pombe*, underscoring the adaptive function of heterochromatin in non-pathogenic yeast [17]. Our findings extend this concept by providing direct evidence that H3K9 methylation contributes to FK506 resistance in dimorphic *Mucor* species. Notably, this mechanism operates in the non-pathogenic species *M. bainieri* and *M. atramentarius*, a species with pathogenic potential, suggesting that heterochromatin-mediated resistance is conserved across diverse fungal taxa, spanning both unicellular yeasts and multinuclear hyphal species. While RNAi-dependent antifungal resistance has been shown to be rapidly induced and reversed following in vivo passage in a murine model [36], the in vivo stability of heterochromatin-dependent epimutants has not been previously characterized. Here, we demonstrate that a heterochromatin-dependent epimutant remains stable in vivo (**Figs 6** and **7**). This persistence, even in the absence of antifungal pressure, raises the possibility that epigenetically resistant strains could silently disseminate in clinical settings, complicating diagnostics and treatment.

In summary, our study extends RNAi-dependent gene silencing to additional *Mucor* species and uncovers that heterochromatin-mediated silencing also drives antifungal resistance. These findings underscore the remarkable plasticity of fungal epigenomes and highlight adaptive strategies that enable reversible, yet durable, resistance. Targeting RNAi and chromatin-modifying pathways may thus represent a promising therapeutic strategy against fAMR.

## Materials and methods

### Ethics statement

All animal experiments in this study were approved by the Duke University Institutional Animal Care and Use Committee (IACUC) (protocol #A224-23-11). Animal care and experiments were conducted according to IACUC ethical guidelines.

## Fungal strains, media, and growth conditions

*Mucor* strains studied here are described in S1 Table. *Mucor* strains were grown on liquid or solid yeast extract peptone dextrose (YPD; BD, Difco, Franklin Lakes, NJ, USA) at room temperature. To isolate FK506-/rapamycin-resistant strains, $10^5$ spores were inoculated onto more than 10 solid YPD plates, supplemented with 1 µg/mL of FK506 (Astellas Pharma, Chuo, Tokyo, Japan) and/or 100 ng/ml of rapamycin (API Chem, Hangzhou, Zhejiang, China) and incubated for 3, 5, 7 days [15]. To test the stability of FK506-resistant strains, spores of the resistant strains were collected and inoculated in liquid YPD without any drugs overnight [16]. These were then serially subcultured overnight for multiple consecutive rounds, after which the passaged cells were spotted onto YPD or YPD supplemented with FK506 to confirm phenotypic plasticity.

## Illumina sequencing and genome assembly

*Mucor* mycelium were grown on YPD medium, harvested, frozen in liquid nitrogen, and ground into a fine powder. Genomic DNA samples were prepared using Norgen Biotek Yeast/Fungi Genomic DNA Isolation Kit (Ontario, Canada). Genomic DNA libraries were constructed with Roche KAPA HyperPrep Kit. 150-bp paired-end reads were obtained utilizing the NovaSeq X Plus sequencing system (Illumina) at the Duke Sequencing and Genomic Technologies Core (SGT). Raw reads were checked for quality metrics using FastQC v0.11.7 and processed by fastp v0.23.4 to remove adapters and low-quality bases or reads. Genomes were assembled using SPAdes v3.15.5 set to isolate default mode. Genome assemblies were cleaned, sorted, masked, and annotated using the Funannotate v1.8.15 [63].

## Nanopore sequencing and genome assembly

A total of $10^6$ spores from *M. bainieri* and *M. atramentarius* WT strains were inoculated into liquid YPD medium and incubated overnight at 25°C. Cells were harvested by centrifugation, washed with distilled water, frozen with liquid nitrogen, and ground with a mortar and pestle. High-molecular-weight (HMW) DNA was extracted from each sample using the Monarch HMW DNA Extraction Kit for Tissue (NEB, Ipswich, MA, USA) according to the manufacturer's protocol. DNA concentration and quality were assessed using a NanoDrop spectrophotometer (ThermoFisher, Waltham, MA, USA) and a Qubit fluorometer with the Qubit dsDNA High Sensitivity kit (ThermoFisher). DNA integrity was confirmed by CHEF electrophoresis performed at 6.0 V in a 1% agarose gel for 18 hours, which revealed intact HMW DNA. Libraries were prepared using the ligation sequencing kit V14 (SQK-LSK114; Oxford Nanopore Technologies, Oxford, United Kingdom) and sequenced on MinION R10.4.1 flow cells (ONT; Oxford Nanopore Technologies) according to the manufacturer's protocol in our laboratory.

POD5 base calling was performed with the Dorado basecaller (v0.5.2; Oxford Nanopore Technologies, https://github.com/nanoporetech/dorado). Demultiplexing and BAM-to-FASTQ conversion were carried out using Dorado demux (v0.5.2). The resulting FASTQ files were error-corrected with Canu (v2.2) and assembled using Flye (v2.9.5-b1801) [64,65]. The expected genome size was set to 50 Mb, and a minimum overlap length of 5 kb was required between reads. The assembled genome was subsequently polished in three iterations using Pilon (v1.24; Broad Institute, https://github.com/broadinstitute/pilon/releases/tag/v1.24) with default parameters utilizing the raw Illumina sequencing data deposited under NCBI BioProject accessions PRJNA1277536 (*M. bainieri* CBS293.63) and PRJNA1277544 (*M. atramentarius* CBS202.28). Repetitive sequences were identified using RepeatMasker (v4.1.5; https://www.repeatmasker.org) with default parameters and genome annotation was performed using Funannotate (v1.8.17) with the masked genome assembly and strain-specific RNA-seq alignments [63]. This pipeline produced high-quality genome assemblies, with *M. bainieri* assembled into 30 contigs (N50 = 6 Mb; 98.7% BUSCO completeness; 9,781 predicted genes) and *M. atramentarius* into 33 contigs (N50 = 2 Mb; 98.8% BUSCO completeness; 11,597 predicted genes).

## The *fkbA* and calcineurin gene sequencing analysis

*Mucor* strains were cultured on YPD medium for three days. Approximately 1–5 mg of harvested cells was suspended in 300 µL Tissue & Cell Lysis Buffer (Lucigen, Middleton, WI, USA) and 1 µL proteinase K (ThermoFisher). The mixture was incubated at 65°C for 45 min, followed by the addition of 150 µL MPC Protein Precipitation Reagent (Lucigen). After centrifugation, the pellet was washed with 70% ethanol and resuspended in distilled water, and genomic DNA (gDNA) quality was analyzed using a Qubit and a NanoDrop (ThermoFisher). The promoter, open reading frame, and terminator regions of the *fkbA* gene were amplified by polymerase chain reaction (PCR) using primers listed in S2 Table. The purified PCR products were sequenced using long-read sequencing with ONT at Plasmidsaurus (Eugene, CA, USA) and analyzed against the reference *fkbA* sequence using Clustal Omega (v1.2.3).

## RNA extraction and quantitative PCR (qPCR) analysis

Total RNA was extracted from 10 mg of *Mucor* cells cultured on YPD, with or without FK506, at room temperature. After overnight lyophilization, each sample was homogenized in a FastPrep bead-beater (MP Biomedicals, Santa Ana, CA, USA) with 0.2 mL of zirconia-silica beads (BioSpec Products, Bartlesville, OK, USA). Total RNA was then purified with the miRNeasy Mini Kit (Qiagen, Hilden, Germany) and RNA quantity and quality were analyzed using a Nanodrop (Thermo Fisher) and a Qubit Fluorometer with a Qubit RNA High Sensitivity kit (ThermoFisher) [66]. The isolated RNA was treated with DNase and subsequently employed for cDNA synthesis with the Maxima H Minus cDNA Synthesis Master Mix (ThermoFisher, Waltham, MA, USA). Real-time PCR was performed with PowerUp SYBR Green Master Mix (ThermoFisher) on a QuantStudio 3 qPCR system (ThermoFisher) in technical triplicate. The primers for qPCR are listed in S2 Table. Gene expression was normalized to *actA*, and relative fold expression was calculated using the $2^{-\Delta\Delta Ct}$ method.

## Small RNA (sRNA) sequencing

sRNA libraries were constructed with the QIAseq miRNA Library Kit (Qiagen) and sequenced on a NovaSeq X platform (Illumina) using 75 bp single-end reads. Raw reads containing adaptor sequence (5′-GTTCAGAGTTCTACAGTCCGAC GATC-3′ and 5′-AACTGTAGGCACCATCAA-3′) were trimmed utilizing BBDuk. Filtered reads were then mapped to the reference genomes (*M. bainieri* CBS293.63; PRJNA1277536 or *M. atramentarius* CBS202.28; PRJNA1277544) using BBMap with default parameters (last modified September 15, 2022) [67]. The resulting SAM files were converted to BAM files, and antisense reads were sorted using SAMtools (v1.21) [68]. BAM files were converted to TDF files using IGV tools with a window size of 20 bp and the mean window function. The data was subsequently visualized in the Integrative Genome Viewer (IGV v2.16.0), where the Y-axis represents the average read coverage per 20-bp window [69]. Peak calling was performed with MACS3 using the --nomodel option with an extension size of 147 bp. sRNA-seq experiments were performed with one biological sample per strain.

## Western blot

Each *Mucor* strain was cultivated in YPD, with or without FK506, at room temperature for three days. Ten mg of *Mucor* cells were lyophilized overnight and homogenized using a FastPrep bead-beater (MP Biomedicals) with 0.2 mL of zirconia-silica beads (BioSpec Products). Pierce RIPA buffer (ThermoFisher) containing a protease inhibitor tablet (Roche) was then added, and each sample was further disrupted with a FastPrep bead-beater. The samples were incubated on ice for at least 30 min. Cell extracts were centrifuged at 14,000 RPM, 4°C for 10 min, and the supernatants were quantified with the Pierce BCA Protein Assay Kit (ThermoFisher). Next, 20 µg of each sample was mixed with Novex Tris-Glycine SDS Sample Buffer (Invitrogen, Waltham, MA, USA) and boiled at 85°C for 5 min. After briefly cooling, the samples were separated by SDS-PAGE, transferred to a PVDF membrane, and incubated overnight at 4°C with an anti-FKBP12 antibody (Invitrogen) or an anti-GAPDH antibody (Proteintech, Rosemont, IL, USA). Following incubation with a horseradish

peroxidase-conjugated anti-rabbit antibody, FKBP12 and GAPDH expression were detected on a ChemiDoc MP imaging system (Bio-Rad, Hercules, CA, USA).

## Chromatin immunoprecipitation (ChIP)-qPCR

A total of $10^6$ spores from each *Mucor* strain were inoculated in liquid YPD medium with or without FK506 and incubated overnight at 25°C. The samples were then cross-linked with 1% formaldehyde (ThermoFisher), followed by addition of glycine (Sigma-Aldrich, St. Louis, MO, USA) at a final concentration of 0.125M. After washing and freezing using liquid nitrogen, each sample was ground with a pestle and suspended in ChIP lysis buffer (50 mM HEPES, pH = 7.5, 150 mM NaCl, 1 mM EDTA, pH = 8, 1% Triton X-100, and 0.1% sodium deoxycholate) containing a protease inhibitor tablet (Roche). Disrupted cells were then sonicated with a Bioruptor (Diagenode) for 40 cycles, with 30 s on and 30 s off and clarified by centrifugation for 5 min. The input samples were stored at −80°C, while the IP samples were incubated overnight at 4°C with Dynabeads Protein A/G (Invitrogen) pre-conjugated with the anti-histone H3K9 dimethyl antibody (Abcam, Cambridge, UK) or the anti-RNA polymerase II antibody (Active Motif, Carlsbad, CA). Crosslinking between protein and DNA in both input and IP samples was reversed with the MAGnify Chromatin Immunoprecipitation System (Invitrogen), and DNA quality and quality were assessed using a Nanodrop (Thermo Fisher) and a Qubit Fluorometer with a Qubit dsDNA High Sensitivity kit (ThermoFisher). qPCR was performed in technical triplicate with primers described in S2 Table. For H3K9me2 ChIP-qPCR, GremLINE1 transposon elements were used as positive controls in *M. atramentarius* [30]. However, our analysis of the *M. bainieri* genome suggests the GremLINE1 elements are absent in the Illumina-based genome assembly. Instead, to develop a positive control for the H3K9me2 ChIP-qPCR analysis in *M. bainieri*, we hypothesized that the loci enriched for H3K9me2 should at least partially overlap with the genomic regions enriched for sRNAs. Specifically, we first identified three genomic loci enriched for sRNAs based on our sRNA analysis and then designed primers targeting these loci and tested them for the presence of H3K9me2 signals using ChIP-qPCR. One of the three loci, *PS3_010814* (identified in the Illumina-based genome assembly deposited under NCBI SRA PRJNA1277536), exhibited consistent H3K9me2 enrichment across samples including WT, epimutants, and revertants, and was selected as the positive control for the subsequent H3K9me2 ChIP-qPCR analyses. The *actA* gene, which lacks both sRNA accumulation and heterochromatin marks, was used as a negative control. H3K9 methylation enrichment was calculated by normalizing the IP DNA to the input DNA with the ΔCt method and further adjusted to the *actA* gene with the ΔΔCt method, followed by $2^{-\Delta\Delta Ct}$ to determine relative enrichment [70,71]. For RNA polymerase II ChIP-qPCR, the *actA* gene was used as a positive control, as it is transcriptionally active. As a negative control, we selected an intergenic locus defined as an siRNA-producing region embedded within transposable element-rich sequences and associated with H3K9me2 heterochromatin. RNA polymerase II enrichment was similarly calculated by using the $2^{-\Delta\Delta Ct}$ method with a negative control, an intronic region enriched for siRNAs and H3K9me2.

## ChIP sequencing

Input and IP libraries were constructed with KAPA HyperPrep (Roche) and sequenced on a NovaSeq X platform (Illumina) using 150 bp paired-end reads. Raw FASTA files were trimmed with Trimmomatic (v0.39) and filtered reads were mapped to the reference genomes (*M. bainieri* CBS293.63; PRJNA1277536 or *M. atramentarius* CBS202.28; PRJNA1277544) using Bowtie2 (v2.5.4) with default parameters [72,73]. The resulting SAM files were converted to BAM files using SAMtools (v1.21) [68]. Broad ChIP-peak calling was analyzed with MACS3 (v3.0.3) with the parameters --mfold 10 1000 and --extsize 147, and enrichment was analyzed by generating coverage files with BamCoverage (v3.5.6), followed by counts per million normalization [74,75]. Processed BAM files were converted to TDF files with IGV tools with a window size of 20 bp and the mean window function and subsequently visualized with IGV (v2.16.0) [76]. ChIP-seq experiments were performed with one biological sample per strain.

### In vivo infection

Six-week-old male BALB/cAnNCrl mice, 6–7 weeks old, obtained from Jackson Laboratory (Bar Harbor, Maine, USA), were analyzed in this study. Five mice were infected with each *Mucor* strain. Two days prior to infection, 200 mg/kg of cyclophosphamide (Sigma-Aldrich), an immunosuppressant, was administered intraperitoneally. Each mouse was infected with $1 \times 10^6$ spores (in 50 μL PBS) via retro-orbital injection after anesthetizing the mice with isoflurane for 2.5 min. After infection, the mice were monitored twice daily and administered cyclophosphamide at 200 mg/kg body weight every four days. To assess pathogenicity, weight loss was monitored, and any animal experiencing ≥20% loss of body weight was humanely euthanized. To examine epimutant stability, five mice per strain were subjected to four treatment conditions: (i) cyclophosphamide at 200 mg/kg every 4 days, (ii) cyclophosphamide at 50 mg/kg every 4 days for 2 weeks, (iii) no cyclophosphamide for 2 weeks, (iv) no cyclophosphamide treatment for 4 weeks. At the designated time points, mice were sacrificed, and fungal burden and epimutant stability were evaluated.

### Stability of epimutants during in vivo infection

To assess fungal burden and isolate *Mucor* strains from organs, five mice per *Mucor* strain were intravenously infected. At the humane endpoint, the brain, lungs, liver, and spleen were dissected. Whole organs were homogenized in 0.8 mL of PBS with a bead beater for 2 min, then appropriate dilutions were plated on YPD supplemented with 50 μg/mL chloramphenicol. After a two-day incubation at room temperature, fungal colonies were picked and patched onto YPD plates with or without FK506 to assess the stability of FK506 resistance following in vivo passage. The stability rate was determined by calculating the proportion of resistant colonies on YPD supplemented with FK506 relative to the total colonies patched on YPD [36]. To determine whether resistant isolates were epimutants, gDNA was extracted from all resistant isolates, and the *fkbA* region was PCR-amplified and sequenced using long-read sequencing with ONT by Plasmidsaurus (Eugene, CA, USA). In addition, to determine whether the resistant isolates represented heterochromatin-associated epimutants, several resistant isolates were analyzed by ChIP-qPCR.

### Quantification and statistical analyses

Statistical differences between the control and target strains were evaluated using ordinary one-way ANOVA followed by Tukey's Honestly Significant Difference test using GraphPad Prism (v10.1.1). Kaplan-Meier survival curves were generated and analyzed with the log-rank test in R (v4.3.2) to estimate survival probability. Data are reported as mean ± standard error of the mean (SEM). In all graphs, *p* values or asterisks indicate statistical significance (*$p < 0.05$; **$p < 0.01$; ***$p < 0.001$; ****$p < 0.0001$).

### Supporting information

**S1 Fig. Isolation of FK506- and rapamycin-resistant strains from the *Mucor circinelloides* complex.** Spores from phylogenetic species (PS) 1 to PS16 were point-inoculated onto YPD plates and incubated at room temperature until resistant colonies emerged. After seven days, resistant colonies were observed in PS1, PS3, PS6, PS10, PS13, PS14, and PS15. PS10, PS14, and PS15 were included as positive controls, as resistant strains had previously been characterized. Red arrows indicate sectors of FK506-resistant growth on the plate. FK506 was at a concentration of 1 μg/mL, while rapamycin was used at a concentration of 100 ng/mL.
(TIFF)

**S2 Fig. Fungal growth and sporangiospore production in *Mucor bainieri* and *Mucor atramentarius*. (A, D)** Point-inoculation phenotypes of *Mucor* strains inoculated with $10^6$ spores and grown on YPD with or without 1 μg/mL FK506 for 4 days. **(B, E)** Quantification of fungal growth for the same strains shown in (A or D). Error bars represent mean ± SEM (*n* = 3). Statistical significance: *$p \leq 0.05$; **$p \leq 0.01$; ***$p \leq 0.001$; ****$p \leq 0.0001$. **(C, F)** Quantification of sporangiospore

production for the same strains shown in (A or D). Error bars represent mean ± SEM ($n = 3$). Statistical significance: *$p \leq 0.05$; **$p \leq 0.01$; ***$p \leq 0.001$; ****$p \leq 0.0001$. WT, wild-type; E1–E9, epimutants; E1P4–E9P24, revertants. The data underlying this figure can be found in S1 Data.
(TIFF)

**S3 Fig. Analysis of 5mC in *Mucor bainieri* and *Mucor atramentarius*. (A)** Presence/absence matrix of 5mC DNA methyltransferases across multiple fungal species. Dark blue indicates the presence of an ortholog, whereas light blue indicates its absence. **(B)** Genome-wide distribution of 5mC in wild-type strain of *Cryptococcus neoformans* Bt210 (positive control), *M. bainieri,* and *M. atramentarius*, detected from Oxford Nanopore sequencing. 5mC was identified using the Dorado basecaller with modified-base models and summarized with modkit pileup under default settings. Nanopore sequencing data of *C. neoformans* Bt210 (published in Huang J and colleagues, *PNAS*, 2024; PMID: 39536081; NCBI BioProject PRJNA1138746) were used as a positive control for the analysis of 5mC in two *Mucor* species.
(TIFF)

**S4 Fig. Small RNAs mediate FK506 resistance in *Mucor bainieri*.** Small RNA coverage mapped across the *fkbA* gene and its neighboring loci. The genomic plot shows a 5 kb region encompassing the *fkbA* locus, including the *fkbA* gene (red box) and the adjacent *patA* gene (*PS3_001330*).
(TIFF)

**S5 Fig. Profiling of small RNAs and relative mRNA abundance for genes neighboring *fkbA* in *Mucor bainieri*.** **(A)** Genomic schematic of the *fkbA* locus and its flanking genes. **(B, E, H)** Antisense sRNA abundance (RPM: reads per million) mapped to genes located upstream and downstream of the *fkbA*. **(C, F, I)** 5′ end nucleotide preference of antisense sRNAs associated with each *fkbA*-neighboring gene. **(D, G, J)** Size distribution of antisense sRNAs mapped to each *fkbA*-neighboring gene. **(K)** Quantification of mRNA expression in genes adjacent to the *fkbA* in WT, epimutants, and revertants. Error bars represent mean ± SEM ($n = 3$). Statistical significance: ****$p \leq 0.0001$. WT, wild-type; E1–E9, epimutants; E1P4–E9P24, revertants. The data underlying this figure can be found in S1 Data.
(TIFF)

**S6 Fig. Small RNAs spread across the *fkbA* locus in *Mucor atramentarius*.** Small RNA coverage mapped across the *fkbA* gene and its neighboring loci. The genomic plot shows a 5 kb region encompassing the *fkbA* locus, including the *fkbA* gene (red box) and the adjacent *patA* gene (*PS6_003788*).
(TIFF)

**S7 Fig. Identification and profiling of small RNAs in genes neighboring *fkbA* in *Mucor atramentarius*. (A)** Genomic schematic of the *fkbA* locus and its flanking genes. **(B, E, H, K, N, Q)** Antisense sRNA abundance (RPM: reads per million) mapped to the *fkbA* gene and its upstream and downstream neighboring genes. **(C, F, I, L, O, R)** 5′ end nucleotide preference of antisense sRNAs associated with the *fkbA* gene and each *fkbA*-neighboring gene. **(D, G, J, M, P, S)** Size distribution of antisense sRNAs mapped to the *fkbA* gene and each *fkbA*-neighboring gene. WT, wild-type; E1 and E2, epimutants; E1P32 and E2P28, revertants. The data underlying this figure can be found in S1 Data.
(TIFF)

**S8 Fig. Relative mRNA abundance of genes neighboring *fkbA* in *Mucor atramentarius*. (A)** Genomic schematic of the *fkbA* locus and its flanking genes. **(B)** Quantification of mRNA expression in genes adjacent to the *fkbA* in WT, epimutants, and revertants. Error bars represent mean ± SEM ($n = 3$). Statistical significance: **$p \leq 0.01$, ***$p \leq 0.001$, and ****$p \leq 0.0001$. WT, wild-type; E1 and E2, epimutants; E1P32 and E2P28, revertants. The data underlying this figure can be found in S1 Data.
(TIFF)

 

**S9 Fig. In vivo fungal burden and *fkbA* sequence profiling of FK506-resistant epimutant derivatives of *Mucor atramentarius*.** **(A)** Fungal burden following retro-orbital inoculation of *M. atramentarius* with cyclophosphamide-treated mice (200 mg/kg) was determined at 4 days post-infection. Fungal burden was expressed as colony-forming units (CFU) per gram of tissue across four organs. Five biological replicates were assessed. Error bars represent mean ± SEM ($n = 5$). WT, wild-type; E1, epimutant; E1P32, revertant. **(B)** Schematic representation of Nanopore sequencing results showing the presence or absence of point mutations across the *fkbA* locus in FK506-resistant colonies recovered from each organ after in vivo infection. PCR amplicons spanning the promoter, open reading frame, and terminator regions of the *fkbA* locus were subjected to long-read Nanopore sequencing, and individual reads were mapped to the reference genome to assess nucleotide variation across the locus. No mutations in the *fkbA* locus were detected. Organ diagrams were generated with BioRender.com. The data underlying this figure can be found in S1 Data.
(TIFF)

**S10 Fig. In vivo fungal burden and chromatin profiling at the *fkbA* locus in *Mucor atramentarius*.** ChIP-qPCR analysis of H3K9me2 enrichment at the *fkbA* locus following in vivo passage, targeting the positive control region, 5′ UTR, and 3′ UTR. Error bars represent mean ± SEM ($n = 3$). Statistical significance: \*\*\*$p ≤ 0.001$; \*\*\*\*$p ≤ 0.0001$. WT, wild-type; E1R1–E1R3, FK506-resistant epimutants; E1S1, FK506-sensitive epimutant; E1P32, revertant. Organ diagrams were generated with BioRender.com. The data underlying this figure can be found in S1 Data.
(TIFF)

**S11 Fig. In vivo fungal burden in immunosuppressed and non-immunosuppressed mice and *fkbA* sequence profiling of FK506-resistant epimutant derivatives of *Mucor atramentarius*.** **(A)** Fungal burden following retro-orbital inoculation of *M. atramentarius* epimutant in cyclophosphamide-treated mice (50 mg/kg) at 2 weeks post-infection. **(B)** Fungal burden in untreated mice at 2 weeks post-infection. **(C)** Fungal burden in untreated mice at 4 weeks post-infection. For all panels, fungal burdens were expressed as colony-forming units (CFU) per gram of tissue across four organs. Five biological replicates were assessed. Error bars represent mean ± SEM ($n = 5$). **(D–F)** Schematic representation of Nanopore sequencing results showing the presence or absence of point mutations across the *fkbA* locus in FK506-resistant colonies recovered from each organ after in vivo infection. PCR amplicons spanning the promoter, open reading frame, and terminator regions of the *fkbA* locus were subjected to long-read Nanopore sequencing, and individual reads were mapped to the reference genome to assess nucleotide variation across the locus. No mutations in the *fkbA* locus were detected. Organ diagrams were generated with BioRender.com. The data underlying this figure can be found in S1 Data.
(TIFF)

**S12 Fig. Comparative distribution of RNAi and heterochromatin-associated proteins across *Mucor* species.** **(A–C)** Proteins that mediate **(A)** RNA interference, **(B)** H3K9 methylation-based heterochromatin formation, and **(C)** RNA-induced transcriptional silencing (RITS) are depicted. Rectangles display the full-length, scaled protein sequences of every identified protein homolog and their predicted, color-coded InterPro protein domains abbreviated as follows: N-Ago (Protein argonaute, N-terminal, IPR032474); L1 (Argonaute, linker 1 domain, IPR014811); PAZ (PAZ domain, IPR003100); L2 (Argonaute linker 2 domain, IPR032472); Mid (Protein argonaute, Mid domain, IPR032473); Piwi (Piwi domain, IPR003165); DEAD/DEAH (DEAD/DEAH box helicase domain, IPR011545); Heli_C (Helicase C-terminal domain-like, IPR001650); DD (Dicer dimerization domain, IPR005034); RNaseIII (Ribonuclease III domain, IPR000999); RRM (RNA recognition motif domain, IPR000504); RDRP (RNA-dependent RNA polymerase, eukaryotic type, IPR007855); RNaseH (Ribonuclease H superfamily, IPR036397); NAM7 (DNA2/NAM7-like helicase, IPR045055); NF-X1 (Zinc finger, NF-X1-type, IPR000967); dsRBD (Double-stranded RNA-binding domain, IPR014720); RING (Zinc finger, RING-type, IPR001841); SET (SET domain, IPR001214); Pre-SET (Pre-SET domain, IPR007728); Post-SET (Post-SET domain,

IPR003616); Cul_NEDD8 (Cullin protein, neddylation domain, IPR019559); Cullin (Cullin homology domain, IPR016158); Wh DBD (Winged helix DNA-binding domain superfamily, IPR036390); CD (Chromo domain, IPR023780); Stc1 (Stc1 domain, IPR024630); RBD (RNA-binding domain superfamily, IPR035979); Chop1 (Chop1, PIN domain, IPR048709); Tas3 (Tas3, C-terminal helical domain, IPR049112); Chp1 CD (Chp1, chromodomain, IPR049937); Chp1 SPOC (Chp1, SPOC domain, IPR049938); Chp1 II (Chp1, domain II, IPR056341). **(D)** A matrix displays the presence (blue) or absence (white) of the proteins described in (a–c) across *M. lusitanicus* PS10, *M. circinelloides* PS15, *M. bainieri* PS3, and *M. atramentarius* PS6, together with *Schizosaccharomyces pombe* as reference. A light shade of blue represents sequence similarity-based homology but lack of functional studies in *S. pombe* (SPAC2C4.08.1). The data underlying this figure can be found in S1 Data.
(TIFF)

**S13 Fig. Genome synteny and epigenetic profiling of regions adjacent to the *fkbA* locus across four *Mucor* species. (A)** Comparative synteny analysis of genomic regions flanking the *fkbA* locus in four *Mucor* species. Homologous genes were initially identified with DIAMOND protein alignment and validated with BLASTP, and synteny blocks were visualized in R. The red asterisk denotes the *fkbA* gene. **(B)** ChIP-seq profiles of H3K9me2 and small RNA coverage across the *fkbA* locus in the four *Mucor* species (*M. bainieri*, *M. atramentarius*, *M. lusitanicus*, and *M. circinelloides*). The red asterisk denotes the *fkbA* gene. Red boxes highlight a region adjacent to *fkbA* that contains a cluster of transposable elements enriched with H3K9me2. H3K9me2 ChIP-seq and small RNA sequencing data for *M. lusitanicus* and *M. circinelloides* were downloaded from NCBI BioProjects PRJNA903107 and PRJNA1170303, respectively, and analyzed with the same pipeline as applied for *M. bainieri* and *M. atramentarius*.
(TIFF)

**S14 Fig. Comparative genomic analysis of GremLINE1 elements across four *Mucor* species.** Genomic views display gene annotations, TE distribution, CEN boundaries, H3K9me2 ChIP enrichment, and sRNA accumulation in the four *Mucor* species (*M. bainieri*, *M. atramentarius*, *M. lusitanicus*, and *M. circinelloides*). GremLINE1 elements, highlighted in red, are present within or near putative centromeres in three *Mucor* species (*M. atramentarius*, *M. lusitanicus*, and *M. circinelloides*), but are absent in the *M. bainieri* genome. Instead, the putative centromeric region of *M. bainieri* is enriched for a distinct repeat fragment, TE00000158. Putative centromere regions were predicted based on the *M. lusitanicus* CEN motif, reported in Navarro-Mendoza and colleagues, *Curr Biol.*, 2019 (PMID: 31679929).
(TIFF)

**S1 Table. *Mucor* strains analyzed in this study.**
(DOCX)

**S2 Table. Oligonucleotides utilized in this study.**
(DOCX)

**S3 Table. *fkbA* sequences from FK506-resistant *Mucor atramentarius* isolates recovered from each organ after 4 days of in vivo passage with cyclophosphamide (200 mg/kg).**
(XLSX)

**S4 Table. *fkbA* sequences from FK506-resistant *Mucor atramentarius* isolates recovered from each organ after 2 weeks of in vivo passage with lower cyclophosphamide (50 mg/kg).**
(XLSX)

**S5 Table. *fkbA* sequences from FK506-resistant *Mucor atramentarius* isolates recovered from each organ after 2 weeks of in vivo passage without cyclophosphamide.**
(XLSX)

**S6 Table.** *fkbA* sequences from FK506-resistant *Mucor atramentarius* isolates recovered from each organ after 4 weeks of in vivo passage without cyclophosphamide.
(XLSX)

**S1 Data. Numerical data underlying all graphs.**
(XLSX)

**S1 Raw Images. Uncropped scans of blots.**
(DOCX)

## Acknowledgments

We thank Dr. Vikas Yadav for his insightful advice on bioinformatics analyses as well as for his scientific discussions and critical reading. We are grateful to Dr. María Isabel Navarro-Mendoza for providing guidance on in vivo training. We also thank Anna Floyd Averette for her invaluable support and assistance with in vivo assays; Ziyan Xu for preparing DNA for Illumina whole genome sequencing; Dr. Jason E. Stajich (University of California) for providing whole-genome sequence datasets and initial preliminary analysis; and Dr. Grit Walther (Hans Knöll Institute) for providing *Mucor* isolates. We appreciate critical reading by Dr. Sheng Sun and Dr. Jun Huang. We thank Devi Swain, Director of the Duke Sequencing and Genomic Technologies Core for technical support.

## Author contributions

**Conceptualization:** Ye-Eun Son, Joseph Heitman.

**Data curation:** Ye-Eun Son.

**Formal analysis:** Ye-Eun Son.

**Funding acquisition:** Joseph Heitman.

**Investigation:** Ye-Eun Son, Carlos Pérez-Arques.

**Methodology:** Ye-Eun Son, Carlos Pérez-Arques.

**Resources:** Joseph Heitman.

**Software:** Ye-Eun Son, Carlos Pérez-Arques.

**Supervision:** Joseph Heitman.

**Validation:** Ye-Eun Son.

**Visualization:** Ye-Eun Son, Carlos Pérez-Arques.

**Writing – original draft:** Ye-Eun Son.

**Writing – review & editing:** Ye-Eun Son, Carlos Pérez-Arques, Joseph Heitman.

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
