## [Editor Report · Decision Letter 0]

26 Nov 2025

Dear Joe,

Thank you for submitting your revised manuscript now entitled "Epimutations driven by RNAi or heterochromatin evoke transient antimicrobial drug resistance in fungi" for consideration as a Research Article by PLOS Biology.

Your manuscript has now been evaluated by the PLOS Biology editorial staff as well as by the previous academic editor and I am writing to let you know that we would like to send your revision back to previous reviewers.

Once your full submission is complete, your paper will undergo a series of checks in preparation for peer review. After your manuscript has passed the checks it will be sent out for review. To provide the metadata for your submission, please Login to Editorial Manager (https://www.editorialmanager.com/pbiology). As I know is currently Thanksgiving overthere, I am giving a week to submit all the required information, by Dec 03 2025 11:59PM.

Best wishes, and a happy Thanksgiving,

Melissa

Melissa Vazquez Hernandez, Ph.D.

Associate Editor

PLOS Biology

---

## [Decision Letter · Decision Letter 1]

18 Dec 2025

Dear Joe,

I hope you are doing great. Thank you for your patience while we considered your revised manuscript "Epimutations driven by RNAi or heterochromatin evoke transient antimicrobial drug resistance in fungi" for publication as a Research Article at PLOS Biology. This revised version of your manuscript has been evaluated by the PLOS Biology editors, the Academic Editor and the original reviewers.

Based on the reviews, we are likely to accept this manuscript for publication, provided you satisfactorily address the remaining points raised by the reviewers, with the exception of any additional transcriptomic analyses, which while could increase the impact, we believe are not necessary. Please also make sure to address the following data and other policy-related requests.

1) We routinely suggest changes to titles to ensure maximum accessibility for a broad, non-specialist readership, and to ensure they reflect the contents of the paper. In this case, we would suggest a minor edit to the title, as follows. Please ensure you change both the manuscript file and the online submission system, as they need to match for final acceptance:

"Epimutations driven by RNAi or heterochromatin evoke transient antimicrobial drug resistance in pathogenic Mucor fungi"

2) Please add the weblink of the funding agencies in the Financial Disclosure statement in the manuscript details.

3) Please add to your Competing Interests the following statement “JH is a member of PLOS Biology’s Editorial Board. The other authors declare that no competing interests exist."

4) The Ethics statement needs to be a separate, independent (and the first) subheading in the Material & Methods section. You currently have it in the subsection "In vivo infection".

Please supply the numerical values either in the a supplementary file or as a permanent DOI’d deposition for the following figures:

Figure 2AE, 3BCD, 4AD, 5BE, 6BCD, 7A-F, S2BCEF, S5B-K, S7B-S, S8B, S9A, S10, S11ABC, S12D

6) Please cite the location of the data clearly in all relevant main and supplementary Figure legends, e.g. “The data underlying this Figure can be found in S1 Data” or “The data underlying this Figure can be found in https://doi.org/10.5281/zenodo.XXXXX”

7) Supplementary files (e.g., excel). Please ensure that all data files are uploaded as 'Supporting Information' and are invariably referred to (in the manuscript, figure legends, and the Description field when uploading your files) using the following format verbatim: S1 Data, S2 Data, etc. Multiple panels of a single or even several figures can be included as multiple sheets in one excel file that is saved using exactly the following convention: S1_Data.xlsx (using an underscore).

8) We require the original, uncropped and minimally adjusted images supporting all blot and gel results reported in the Figures 2CF

-- We will require these files before a manuscript can be accepted so please prepare and upload them now. Please carefully read our guidelines for how to prepare and upload this data: https://journals.plos.org/plosbiology/s/figures#loc-blot-and-gel-reporting-requirements

9) Please ensure that your Data Statement in the submission system accurately describes where your data can be found and is in final format, as it will be published as written there

We expect to receive your revised manuscript within four weeks.

*Published Peer Review History*

*Press*

As a reminder, PLOS Biology will be "closed" from Dec 22 to Jan 3, so all actions will be slow during this period. I would like to take the chance to also wish you, your family and your team, the best on these coming holidays and a happy new year.

Sincerely,

Melissa

Melissa Vazquez Hernandez, Ph.D.

Associate Editor

PLOS Biology

REVIEWERS' COMMENTS:

Reviewer #1:

I have carefully evaluated the revised manuscript and the detailed point-by-point responses provided by the authors. I am satisfied that all major concerns raised in the previous review round have been thoroughly and adequately addressed. The authors have significantly strengthened the manuscript by providing new genomic and methylation analyses, clarifying resistant isolate screening, and generating fully documented high-quality reference genomes that support all downstream sequencing work. They also improved the rigor of the ChIP and sRNA analyses, extended and validated the in vivo experiments to rule out genetic resistance, and enhanced overall methodological transparency and contextual framing. Collectively, these revisions fully address the concerns raised and considerably improve the clarity and scientific robustness of the study.

Just a few minor comments about typos found during the new reading:

"find powder" → should be "fine powder"

Occurs in the Illumina sequencing methods description.

"Chian" → should be "China"

In the description of rapamycin source ("Zhejiang, Chian").

Sentence fragment or missing word in "One of the three loc,"

"loc" is an incomplete word; likely intended as "loci."

Inconsistent gene name formatting ("H3K9m2" vs. "H3K9me2")

At least one instance uses "H3K9m2," missing the "e."

"ground into a find powder" appears twice

First in Illumina prep, again in Nanopore prep—both should be "fine powder."

"following to the manufacturer's protocol" → should be "according to the manufacturer's protocol"

Grammatically incorrect phrasing in Nanopore sequencing workflow.

"These were then serially subcultured overnight for several rounds"

Awkward and ambiguous wording; unclear subject and duration.

Reviewer #2:

I really appreciate the authors detailed responses to my questions. In particular, I want to highlight the tremendous value that I thought the analysis of RNAi/CLRC/RITS subunits added to strengthen the manuscript and also the comparative genomic analysis of GremLINES. It is very interesting that these elements are missing in M.bainieri. These analyses clearly highlight the power of comparative genomic analysis, reveal important differences across Mucor species and at least provide opportunities to envision possibilities for why RNAi and H3K9me show such wide mechanistic disparities during adaptation. The only concern I have is that the experiments are clearly an opening for comparative transcriptomic analysis. The authors state pursuing transcriptomic analysis to identify adaptation mechanisms in future studies. Upon reading the reviewer responses, I am still left thinking that the transcriptomic data would be a welcome addition to the current manuscript. This is my only concern because it is also an opportunity to set this manuscript apart from earlier work and really use the power of comparative analysis across species. That being said, I am still a fan of the paper. The additional experiments, the extended discussion are all welcome additions that make the paper a very engaging read.

Reviewer #3:

The revised version contains further analysis and discussion that address my comments convincingly

---

## [Editor Report · Decision Letter 2]

5 Jan 2026

Dear Joe,

Happy New Year!

I hope you are doing great and having a nice start into 2026. Thank you for the submission of your revised Research Article "Epimutations driven by RNAi or heterochromatin evoke transient antimicrobial drug resistance in pathogenic Mucor fungi" for publication in PLOS Biology. On behalf of my colleagues and the Academic Editor, Aaron P. Mitchell, I am pleased to say that we can in principle accept your manuscript for publication, provided you address any remaining formatting and reporting issues. These will be detailed in an email you should receive within 2-3 business days from our colleagues in the journal operations team; no action is required from you until then. Please note that we will not be able to formally accept your manuscript and schedule it for publication until you have completed any requested changes.

IMPORTANT: Many thanks for now providing an independent Ethics Statement. However, we require that the Ethics Statement is places as the first subheading in Materials and Methods. Currently, you have it at the end. Could you please move it? I have asked my colleagues to include this request alongside their own.

PRESS

Sincerely,

Melissa

Melissa Vazquez Hernandez, Ph.D., Ph.D.

Associate Editor

PLOS Biology
